# SimSurvey: An R package for comparing the design and analysis of surveys by simulating spatially-correlated populations

**Paul M. Regular**\*, **Gregory J. Robertson**¤, **Keith P. Lewis, Jonathan Babyn, Brian Healey, Fran Mowbray**

Fisheries and Oceans Canada, Northwest Atlantic Fisheries Center, St. John's, Newfoundland and Labrador, Canada

¤ Current address: Environment and Climate Change Canada, Mount Pearl, Newfoundland and Labrador, Canada

\* Paul.Regular@dfo-mpo.gc.ca

## Abstract

Populations often show complex spatial and temporal dynamics, creating challenges in designing and implementing effective surveys. Inappropriate sampling designs can potentially lead to both under-sampling (reducing precision) and over-sampling (through the extensive and potentially expensive sampling of correlated metrics). These issues can be difficult to identify and avoid in sample surveys of fish populations as they tend to be costly and comprised of multiple levels of sampling. Population estimates are therefore affected by each level of sampling as well as the pathway taken to analyze such data. Though simulations are a useful tool for exploring the efficacy of specific sampling strategies and statistical methods, there are a limited number of tools that facilitate the simulation testing of a range of sampling and analytical pathways for multi-stage survey data. Here we introduce the R package **SimSurvey**, which has been designed to simplify the process of simulating surveys of age-structured and spatially-distributed populations. The package allows the user to simulate age-structured populations that vary in space and time and explore the efficacy of a range of built-in or user-defined sampling protocols to reproduce the population parameters of the known population. **SimSurvey** also includes a function for estimating the stratified mean and variance of the population from the simulated survey data. We demonstrate the use of this package using a case study and show that it can reveal unexpected sources of bias and be used to explore design-based solutions to such problems. In summary, **SimSurvey** can serve as a convenient, accessible and flexible platform for simulating a wide range of sampling strategies for fish stocks and other populations that show complex structuring. Various statistical approaches can then be applied to the results to test the efficacy of different analytical approaches.

## Introduction

Good survey design is a critical prerequisite for obtaining accurate and precise estimates of a population of interest. The need for sound sampling techniques is as relevant today as it was in

**Data Availability Statement:** The simulated data used in this manuscript can be replicated using the code provided in the S1 Appendix.

**Funding:** This work was supported by the NSERC visiting-fellow program and Fisheries and Oceans Canada. The Science Branch of Fisheries and Oceans Canada initiated this project to assess the sampling design of trawl surveys conducted along the Newfoundland and Labrador shelf.

**Competing interests:** The authors have declared that no competing interests exist.

the 1930's when much of the theory and principles behind the collection of scientific data were developed by statisticians such as R.A. Fisher and Jerzy Neyman [1]. Then and now, we are experiencing a period of increasing human activity and robust methods are required to detect the impacts of our actions on the natural world [2]. Survey quality is especially important for conservation-oriented science as monitoring data are used to identify species at risk of extinction [3], to assess the efficacy of recovery plans [4], and to determine precautionary levels of exploitation [5]. Surveys with a solid basis in sampling theory have therefore become a core component of many ecological monitoring programs throughout the world.

One of the biggest challenges associated with surveys is the cost of planning and implementing effective and efficient sampling designs [6]. Despite their costs, surveys have become a mainstay in the management of dynamic fish stocks as survey data are not easily replaced by other sources of information [7]. For instance, large population declines can be overlooked by relying solely on catch and effort data from commercial fisheries [8]. Such difficulties have contributed to the growing reliance on fisheries-independent surveys to monitor fish populations. Planning or improving such surveys is an important but non-trivial task because of the inherent complexity of both the survey and target population. Specifically, fisheries-independent surveys usually involve multiple levels of sampling and fish populations often show size-structured spatial and temporal dynamics [9]. These features are not easily captured by simple statistical models, which makes it difficult to determine optimal sampling and sub-sampling schemes [10–12]. In such situations it is common for practitioners to resort to simulations to test various sampling and analytical pathways (e.g. [13]). However, such tests are rare in fisheries research (but see [14,15]), perhaps because of the complexities associated with fisheries-independent surveys.

Here we document **SimSurvey**, an R package designed to facilitate realistic simulations of fisheries-independent trawl surveys. In short, the package allows for the simulation of random or stratified-random surveys of an age-structured population that varies in space and time. The package has two main components: the first focuses on mimicking realistic fish stocks by simulating a spatially and age-correlated population distributed across a habitat gradient, and the second component focuses on simulating various surveys of these virtual fish stocks. Although constructed to assess the design of fisheries-independent trawl surveys, this package can be used to model any population that shows complex spatial and age structure.

This simulation framework has similarities to those presented by Schnute and Haigh [14] and Puerta et al. [15], however, we focused our efforts on developing a series of general and accessible functions to simplify the process of testing multiple sampling scenarios and analytical pathways. The steps taken to simulate surveys of spatial, age-structure populations are outlined below. We first outline the equations underlying the package (**Model structure** section) and then we demonstrate how to use its core functions (**Core functions** section). The core functions of the package are largely demonstrated using default settings, and these settings are based on a case study (see S1 Appendix for details on the case study, and see S2 Appendix for guidance on how to modify default settings to suit specific needs). Several of the results from the case study are described and discussed in the **Interpretation** section as they highlight one use case of the package. Finally, we discuss the broader research opportunities and future directions of the package (Discussion section).

## Model structure

In this section, we describe the framework currently implemented in the **SimSurvey** package. With this framework, we tried to strike a balance between realism, simplicity, generality and computational feasibility. The framework follows four general steps: 1) simulate a spatially-aggregated age-structured population; 2) distribute the population throughout a spatial grid,

imposing correlation across space, time and age; 3) sample the population using random sampling; and, 4) obtain population estimates using a design-based analysis. Though there is a degree of flexibility in each of these steps, users can circumvent specific components by applying user defined equations, inputs and/or analyses. Details on how to use the package and, if desired, circumvent some aspects of its structure are outlined in the **Core functions** section.

## Simulate abundance

The simulation starts with an exponential decay cohort model where the abundance at age $a$ in year $y$ ($N_{a,y}$) equals the abundance of that cohort in the previous year multiplied by the associated survival rate, which is expressed in terms of total mortality ($Z_{a,y}$):

$$N_{a,y} = N_{a-1,y-1}e^{-Z_{a-1,y-1}}$$

Here, numbers at age in the first year are filled via exponential decay, $N_{a,1} = N_{a-1,1}e^{-Z_{a-1,1}}$, numbers at age 1 (i.e. recruits) vary around a baseline value, $log(N_{1,y}) = log(\mu_r) + \epsilon_y$, and total mortality is set to a baseline level plus process error, $log(Z_{a,y}) = log(\mu_Z) + \delta_{a,y}$. The error around the recruitment process was set to follow a random walk, $\epsilon_y \sim N(\epsilon_{y-1}, \sigma_r^2)$, and the process error was simulated using the covariance structure described in Cadigan [16], $\delta_{a,y} \sim N(0,\Sigma_{a,y})$. The covariance across ages and years is controlled by a process error variance parameter ($\sigma_\delta^2$) along with age and year correlation parameters ($\varphi_{\delta,\text{age}}$ and $\varphi_{\delta,\text{year}}$, respectively). This structure allows for autocorrelation in process errors across ages and years (i.e. total mortality can be made to be more similar for fish that are closer together in age and/or time). Note that a plus group is not modeled as the number of ages can easily be extended to include groups with zero fish.

In practice, abundance at age is often inferred from length data as it is easier to collect. Abundance at length is therefore simulated from abundance at age using the original von Bertalanffy growth curve [17]:

$$log(L) = log(L_\infty - (L_\infty - L_0)e^{-Ka}) + \varepsilon$$

Where $L_\infty$ is the mean asymptotic length, $L_0$ is length at birth, $K$ is the growth rate parameter and the error is assumed to follow the normal distribution, $\varepsilon \sim N(0, \sigma_L^2)$. Numbers at age are distributed across discrete length groups following a lognormal distribution by calculating the probability of being in a specific length group given age, $\phi_{a,l}$. These probabilities are calculated using the standard normal cumulative density function $\Phi$ for a sequence of length groups $l$ from length 0 to 10 times the maximum predicted length $L$ at an interval of $l_{group}^N$:

$$\phi_{a,l} = \Phi\left(\frac{log(L_l)}{\sigma_L}\right) - \Phi\left(\frac{log(L_{l-1})}{\sigma_L}\right)$$

Overall, this formulation facilitates the simulation of a dynamic length and age structured population. Though some typical relationships have yet to be implemented (e.g. stock-recruitment), sufficient information can be simulated to assess survey performance across a range of abundance levels across years, lengths and ages.

## Simulate spatial distribution

Rather than developing a full spatially-explicit model, population and spatial dynamics are modeled as independent processes for simplicity. The complexities of spatial population dynamics—such as larval dispersal, spatial differences in growth and population connectivity—are not explicitly accounted for and, as such, the model is a necessary simplification of reality. Despite this limitation, the approach taken facilitates the simulation of spatial, age-structured

populations with sufficient complexity for testing the efficacy of various survey designs. The simplicity also limits the number of unknown parameters that need to be specified to simulate a population. Parameter estimates from spatially-aggregated age-structured models, which are commonly used in stock assessments, can therefore be used to simulate a population using the cohort model and the resultant abundance at age values can be distributed across a spatial grid. Here, a grid of $s$ cells is generated where each cell has an area of $S$ and depth $d$; depth is defined using a sigmoid curve, applied across one spatial axis, with a depth range of $[d_{\min}, d_{\max}]$, shelf depth of $d_{\text{shelf}}$ and a shelf width of $w_{\text{shelf}}$. We use depth as our main stratification variable, but note that any other appropriate stratification variables could be used. The grid can be divided into two hierarchical levels, management divisions and habitat-based survey strata. For demonstration purposes, we envision these levels as part of a stratified-random survey within international fishery divisions, i.e., $H_{\text{strat}}$ depth-based strata within $H_{\text{div}}$ divisions (e.g. NAFO or ICES divisions, or any other geographically bounded area). The simulated population is distributed through the grid by simulating spatial-temporal noise controlled by a parabolic relationship with depth and covariance between ages, years and space. This noise term, $\eta_{a,y,s}$, is scaled to sum to 1 to ensure that the total population of each age for each year through the grid equals the number simulated by the cohort model:

$$N_{a,y,s} = N_{a,y} \frac{\eta_{a,y,s}}{\sum_{s=1}^{N_s} \eta_{a,y,s}}$$

$$\eta_{a,y,s} = \frac{(d_s - \mu_d)^2}{2\sigma_d^2} + \xi_{a,y,s}$$

Where $d_s$ is the depth in a specific cell of the grid, $\mu_d$ is the mean depth where abundance is typically highest and $\sigma_d$ controls the width or dispersion of abundance around the mean depth. Residual noise $\xi_{a,y,s}$ is added to this depth relationship using a combination of Matérn covariance, to control the level of spatial aggregation within ages and years, and a two dimension AR1 age-year covariance described in Cadigan [16], to control the level of similarity in distributions across ages and years. The rate at which point-to-point spatial correlation decays with distance is controlled by a smoothing ($\lambda$) and a scaling parameter ($\kappa$) (here $\kappa$ is approximated from range parameter $r$, $\kappa = \sqrt{8\lambda}/r$; [18]) and correlation across ages and years is controlled by $\varphi_{\xi,\text{age}}$ and $\varphi_{\xi,\text{year}}$, respectively. The overall variance of the spatial process is controlled by $\sigma_\xi^2$ (see S3 Appendix for a more detailed description of the space-age-year covariance structure). In short, this formulation allows control of depth preferences, the level of spatial aggregation and the degree of age and year specific clustering.

## Simulate survey

The final step in the simulation is to sample the simulated population over the age-year-space array generated. The sampling is random or stratified random, emulating surveys conducted by many research institutions around the world. The area of each strata $A_{\text{strat}}$ is calculated and this is used to define the number of sampling stations $H_{\text{sets}}$, hereafter referred to as sets, allocated to one or more strata under a particular set density, $D_{\text{sets}}$ (i.e. $H_{\text{sets}} = A_{\text{strat}} D_{\text{sets}}$). The allocated number of cells are randomly selected in each strata and the number of fish caught in each set is calculated by applying binomial sampling of the fish in each sampled cell by the proportion of the area covered by the trawl and the catchability of each age:

$$n_{a,y,s} \sim Bin\left(N_{a,y,s}, \frac{A_{\text{trawl}}}{A_{\text{cell}}} q_a\right)$$

Where $n_{a,y,s}$ is the number of fish of age $a$ in year $y$ sampled by a set at location $s$, $A_{\text{trawl}}$ indicates the area covered by the trawl, $A_{\text{cell}}$ is the area of a grid cell, and $q_a$ is the catchability coefficient of each age (i.e. the ability of the trawling gear to catch specific age groups). Here, catchabilities were defined using a logistic curve controlled by a steepness, $k$, and midpoint parameter, $x_0$. In cases where there are multiple sets in one cell, the population in that cell is divided across the sets. While this means that numbers caught in an isolated simulation cannot exceed the numbers in the population, keep in mind that the survey, no matter how intense, is assumed to have no impact on the population from one year to the next.

Once catches are simulated, lengths of the fish sampled by each set are simulated using the von Bertalanffy growth equation found above in the **Simulate abundance** section. Sub-sampling is then conducted whereby a subset of fish are sampled for length measurements and a subset of this subset are sampled for age determination. Specifically, a maximum number of lengths are measured per set ($M_{\text{lengths}}$) and a maximum number of ages ($M_{\text{ages}}$) are sampled per length group ($l_{\text{group}}$) per division, strata or set ($s_{\text{group}}$). Such sub-sampling is common in fisheries-independent surveys as it is costly, impractical and unnecessary to sample every fish captured. Age determination is especially time-consuming, which is why otoliths for age-determination tend to be sub-sampled by length-bin to obtain a representative age sample across a wider range of lengths than would be obtained via random sampling.

## Stratified analysis

While there are many model-based options for obtaining an abundance index from survey data (e.g. [19]), design-based approaches, such as stratified analyses, are often used. Here we apply formulae presented in Smith and Somerton [20] (equations are replicated in S4 Appendix) to calculate year $y$ and simulation $j$ specific stratified estimates of total abundance ($\hat{I}_{y,j}$), abundance at length ($\hat{I}_{l,y,j}$) and abundance at age ($\hat{I}_{a,y,j}$). Note that estimates of total abundance are based on total numbers caught at each set, $n_i$, while abundance at length requires the sub-sampled length frequencies at each set, $m_{l,i}$, to be scaled up using set-specific ratios of the number of fish measured, $m_i$, to numbers caught, $n_i$:

$$n_{l,i} = m_{l,i} \times \frac{n_i}{m_i}$$

Likewise, age frequencies need to be calculated to obtain stratified estimates of abundance at age. This is done by constructing an age-length key, which is the proportion of fish in each length bin that fall into specific age classes. Once these proportions are calculated, they are applied to the bumped up length frequencies, $n_{l,i}$, to approximate age frequencies, $n_{a,i}$, that is:

$$n_{a,i} = \sum_{l=1}^{L} n_{l,i} \times \frac{m'_{a,l,g}}{m'_{l,g}}$$

Here, the prime symbol $'$ indicates that these values are tertiary sampling units as they represent fish sub-sampled for age determination from those sub-sampled for length measurements. Within this sub-sample, $m'_{l,g}$ represents the number of fish within length bin $l$ and from area $g$, and $m'_{a,l,g}$ represents the number of fish within age-class $a$, length bin $l$ and area $g$. The area $g$ can be division, strata or set specific. Though age-length keys are typically constructed and applied over large spatial scales (e.g. division), Aanes and Vølstad [21] recommends calculating proportions at finer scales to better account for hierarchical sample designs.

We used root-mean-squared error (RMSE) as a measure of the precision and bias of the abundance at age estimates from each survey:

$$RMSE = \sqrt{\frac{\sum_{a=1}^{A} \sum_{y=1}^{Y} \sum_{j=1}^{J} (\hat{I}_{a,y,j} - I_{a,y})^2}{A \times Y \times J}}$$

Where $A$, $Y$, and $J$ are the number of ages, years and simulations, respectively, and $I_{a,y}$ is the true abundance available to the survey (i.e. catchability corrected abundance; $I_{a,y} = q_a N_{a,y}$). RMSE was also calculated for abundance at length estimates, where the above formula is indexed by length groups $l$, and total abundance, which lacks a group index of $a$ or $l$.

## Core functions

The **SimSurvey** package was written in the programming language R [22] and it holds a series of functions for 1) simulating the abundance and distribution of virtual fish populations with correlation across space, time and age (sim_abundance, sim_distribution), 2) simulating surveys with a range of sampling strategies and intensities (sim_survey), and 3) estimating the stratified mean and variance of simulated survey data (run_strat; Table 1). **SimSurvey** relies heavily on functions from the **data.table** [23], **raster** [24] and **plotly** [25] packages for their efficient data processing, geographic and plotting facilities, respectively. Package documentation has been published online using **pkgdown** (https://paulregular.github.io/SimSurvey/) and all the source R code behind **SimSurvey** is available on GitHub (https://github.com/PaulRegular/SimSurvey). **SimSurvey** can be installed via GitHub using the **remotes** package:

```
install.packages("remotes")

remotes::install_github("PaulRegular/SimSurvey")
```

The equations behind the functions listed in Table 1 are detailed in the **Model structure** section. Note that several of the core equations are implemented using "closures", which are

**Table 1. Names and descriptions of the key functions of SimSurvey.** Functions in bold font are core functions and those in medium font are designed for use inside the core functions. The latter are typically closures, which are functions that contain data and return functions [26]; here they are used to store parameter values and return functions that require dimensions, such as ages or years, to be supplied.

| Function | Description |
|---|---|
| **sim_abundance** | Simulate a basic age-structured population dynamics model |
| sim_R, sim_Z, sim_N0, sim_vonB | Closures, to use inside sim_abundance, for simulating recruitment, total mortality, initial abundance and growth, respectively |
| **sim_distribution** | Simulate spatial and temporal distribution of an age-structured population |
| sim_ays_covar, sim_parabola | Closures, to use inside sim_distribution, for simulating age-year-space covariance and parabolic relationships with covariates (e.g. depth), respectively |
| make_grid | Make a basic depth stratified square grid to use inside sim_distribution |
| **sim_survey** | Simulate a survey of a spatial, age-structured population |
| sim_logistic | Closure, to use inside sim_survey, for simulating age-specific catchability as a logistic curve |
| **run_strat** | Run a stratified analysis on simulated survey data |
| **strat_error** | Calculate the error of stratified estimates (e.g. root mean squared error of stratified estimates from true values) |
| **test_surveys** | Test the sampling design of multiple surveys using a stratified analysis (internally loops over sim_survey, run_strat and strat_error) |
| expand_surveys | Create a data frame, for use in test_surveys, with all combinations of supplied survey settings |

functions that contain data and return functions [26]. For example, sim_R returns a function that holds the supplied parameter values and requires a sequence of years to be supplied.

```
R_fun <- sim_R(log_mean = log(500), log_sd = 0.5)

R_vec <- R_fun(years = 1:100)
```

Here, the R_vec object holds 100 years of simulated recruitment values and each run of the R_fun function will result in different simulated values using the internal formulation and the parameters supplied to sim_R. The other closures included in the package operate in a similar way in that parameter inputs are supplied to the closure and the functions returned by the closure requires inputs such as ages and/or years. This was done to avoid the repeated specifications of key arguments, such as ages and years. Moreover, this approach provides an option for advanced R users to inspect and modify the closures implemented in the package to supply custom closures with alternate equations (see https://paulregular.github.io/SimSurvey/articles/custom_closures.html for a short vignette on creating custom closures). Also note that each of the closures implemented in the package includes a plot argument such that quick visuals can be obtained using a line of code like this: sim_R(log_mean = log(500), log_sd = 0.5, plot = TRUE)(years = 1:100)).

## sim_abundance

Abundance at age and length is simulated using the sim_abundance function and a default function call is described in Table 2 along with associated symbols from the equations outlined in the **Simulate abundance** section. This function has a simple structure and requires the specification of a series of ages and years along with a series of closures, such as sim_R, sim_Z and sim_vonB, for simulating recruitment (R), total mortality (Z) and growth (growth), respectively. Overall, the function provides a simple tool for simulating a range of dynamic age-structured populations. For instance, below we provide examples where we simulate a relatively long and short lived species (note that default variance, starting abundance and growth settings were used in both simulations).

```
set.seed(438)

long <- sim_abundance(ages = 1:20,

                        R = sim_R(log_mean = log(3e+07)),

                        Z = sim_Z(log_mean = log(0.2)))

short <- sim_abundance(ages = 1:6,

                         R = sim_R(log_mean = log(1e+10)),

                         Z = sim_Z(log_mean = log(0.8)))
```

**Table 2. Default sim_abundance function call, with descriptions, default values and associated parameter symbols of key arguments.**

| Function call | Description | Symbol |
|---|---|---|
| sim_abundance( | | |
| ages = 1:20, | Ages | $a$ |
| years = 1:20, | Years | $y$ |
| R = sim_R(log_mean = log(30000000), | Mean log recruitment[1] | $\mu_r$ |
| log_sd = 0.5), | Standard deviation of log recruitment | $\sigma_r$ |
| Z = sim_Z(log_mean = log(0.5), | Mean log total mortality[2] | $\mu_Z$ |
| log_sd = 0.2, | Standard deviation of log total mortality | $\sigma_Z$ |
| phi_age = 0.9, | Correlation across ages in error around total mortality | $\varphi_{\delta,\text{age}}$ |
| phi_year = 0.5), | Correlation across years in error around total mortality | $\varphi_{\delta,\text{year}}$ |
| growth = sim_vonB(Linf = 120, | Mean asymptotic length (cm) | $L_\infty$ |
| L0 = 5, | Length in birth year (cm) | $L_0$ |
| K = 0.1, | Growth rate | $K$ |
| log_sd = 0.1, | Standard deviation of the von Bertalanffy growth curve | $\sigma_L$ |
| length_group = 3)) | Length group bin size for abundance at length (cm) | $l^N_{\text{group}}$ |

[1] Can be a vector of means with a length equal to the number of years in the simulation.

[2] Can be a matrix of means with number of rows and columns equaling the number of ages and years in the simulation, respectively.

The sim_abundance function returns a list with the sequence of ages (ages), sequence of years (years), sequence of lengths (lengths), numbers of recruits across all years (R), numbers at age in the first year (N0), total mortality matrix (Z), abundance at age matrix (N), abundance at length matrix (N_at_length) and the function supplied to the growth argument (sim_length). The growth function is retained for later use in sim_survey to simulate lengths given simulated catch at age in a simulated survey.

The package also includes several plotting functions for making quick plotly-based [25] interactive visuals of the simulated population. For instance, the plot_surface function can be used to make quick visuals of matrices contained within the list returned by sim_abundance. As an example, we display the abundance at age matrix (object named N in the list produced by sim_abundance; Fig 1); other names can be supplied to the mat argument to visualize a different matrix from the sim_abundance list, such as Z.

```
plot_surface(long, mat = "N")

plot_surface(short, mat = "N")
```

## sim_distribution

The equations outlined in the **Simulate spatial distribution** section are used in the make_grid, sim_ays_covar and sim_parabola functions, and these functions are used within sim_distribution to distribute a population simulated using sim_abundance throughout a grid (Table 3). The output from make_grid is a raster object [24] with four layers: depth, cell, division and

strat. If a more detailed and realistic grid is required, users can manually generate their own survey grid using real data and this grid can be supplied as a raster to sim_distribution if the same structure and correct projection is used. The package includes a manually constructed survey grid of NAFO Subdivision 3Ps off the southern coast of Newfoundland (named survey_grid) and the data-raw folder in the GitHub directory includes the data and code used to construct this grid. However, for simplicity, we use make_grid to construct a square grid for a default run of sim_distribution. Below we generate and plot (Fig 2) a default grid, another grid with the number of strata increased by increasing the number of strat_splits, and another with the number of divisions increased using n_div and a linear depth gradient (the sigmoid curve is forced to be linear when shelf_width is set to zero).

```r
a <- make_grid(n_div = 1, strat_splits = 2, shelf_depth = 200,

               shelf_width = 100, depth_range = c(0, 1000))

b <- make_grid(n_div = 1, strat_splits = 3, shelf_depth = 200,

               shelf_width = 100, depth_range = c(0, 1000))

c <- make_grid(n_div = 4, strat_splits = 1, shelf_depth = 500,

               shelf_width = 0, depth_range = c(0, 1000))

plot_grid(a)

plot_grid(b)

plot_grid(c)
```

   In addition to supplying objects produced by sim_abundance and make_grid, the sim_distribution function requires two closures that describe the age-year-space covariance and the relationship with depth. Here we use sim_ays_covar and sim_parabola to control these relationships and a wide range of age and year specific distributions can be obtained by adjusting a few parameters in these closures (Fig 3). While here we only describe one option for simulating a spatial noise (see S3 Appendix for details), custom closures can be used that leverage simulation models provided by packages such as **RandomFields** [27] or **INLA** [28] (see the code behind sim_ays_covar_sped [https://github.com/PaulRegular/SimSurvey/blob/master/R/sim_dist_spde.R] for an example of how the sim_ays_covar closure was modified to apply a Stochastic Partial Differential Equation approach using the **INLA** package). Below we run a default sim_distribution call, which generates a population that forms tight clusters that are more strongly correlated across years than ages, and another call that generates a population that is more diffuse (i.e. wider range) and exhibits stronger correlation across ages than years

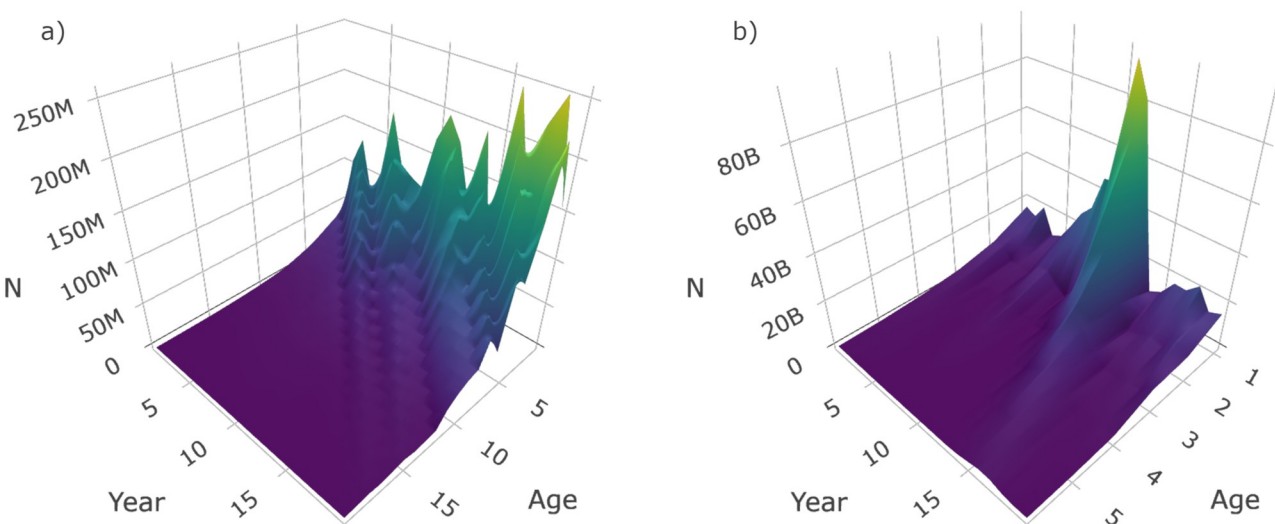

**Fig 1. Surface plots of simulated abundance at age of a relatively a) long lived and b) short lived species.** These plots were produced by plot_surface when supplied a list produced by sim_abundance.

**Table 3. Default sim_distribution function call, with descriptions and associated parameter symbols of key arguments.**

| Function call | Description | Symbol |
|---|---|---|
| sim_distribution( | | |
| sim, | Simulated population from sim_abundance | |
| grid = make_grid(x_range = c(-140, 140), | Range of grid in the x dimension | |
| y_range = c(-140, 140), | Range of grid in the y dimension | |
| res = c(3.5, 3.5), | Grid resolution in x and y dimensions (km)—i.e. cell area | $A_{cell}$ |
| shelf_depth = 200, | Shelf depth (m) | $d_{shelf}$ |
| shelf_width = 100, | Shelf width (km) | $w_{shelf}$ |
| depth_range = c(0, 1000), | Depth range from coast to slope (m) | $[d_{min}, d_{max}]$ |
| n_div = 1, | Number of divisions | $H_{div}$ |
| strat_splits = 2, | Number of strata within each depth class | |
| strat_breaks = seq(0, 1000, 40)), | Series of depth breaks for defining strata | |
| ays_covar = sim_ays_covar(sd = 2.8, | Standard deviation of age-year-space distribution | $\sigma_\xi$ |
| range = 300, | Range of spatial correlation (km) | $r$ |
| lambda = 1, | Smoothness of spatial correlation | $\lambda$ |
| phi_age = 0.5, | Correlation across ages in spatial distribution | $\varphi_{\xi,age}$ |
| phi_year = 0.9, | Correlation across years in spatial distribution | $\varphi_{\xi,year}$ |
| group_ages = 5:20, | Make space-age-year noise equal across these ages[1] | |
| group_years = NULL), | Make space-age-year noise equal across these years[1] | |
| depth_par = sim_parabola(mu = 200, | Depth at which abundance is typically highest (m) | $\mu_d$ |
| sigma = 70)) | Dispersion around depth of peak abundance (m) | $\sigma_d$ |

[1] All ages or years are independent if argument values is NULL.

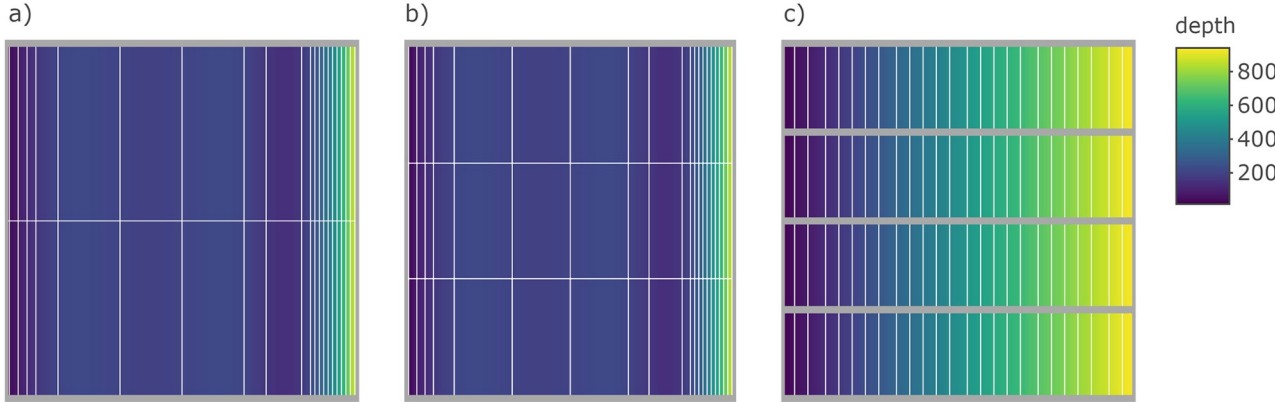

**Fig 2. Plots produced by plot_grid when supplied a raster object produced by make_grid a) using default settings, b) settings that increase the number of times depth strata are split (strat_splits = 3), and c) settings that produce a more linear depth gradient (shelf_width = 0) and increase the number of divisions (n_div = 4).** In these plots, the yellow to purple color gradient represents depth, the thick grey lines delineate divisions and thin white lines delineate strata.

(i.e. lower phi_year and higher phi_age). Distributions can also be forced to be the same across ages and years by using the group_ages and group_years arguments, respectively, in the sim_ays_covar closure. Variance in the size of the clusters can also be modified by changing the sd argument in the sim_ays_covar function. In other words, these parameters can be modified to control the degree of age-specific clustering and inter-annual site-fidelity exhibited by the simulated population. Note that the resolution of the default grid is high and, as such, the simulations below may take minutes to complete. Also note that the key functions in the **Sim-Survey** package have been set-up to be pipe (%>%; [29]) friendly such that values from one function call are forwarded to the next function (i.e. output from the two calls below are functionally the same though the approach is slightly different).

```
set.seed(438)

a <- sim_distribution(sim = sim_abundance(), # nested approach

                      ays_covar = sim_ays_covar(range = 300, # clustered

                                                phi_year = 0.9,

                                                phi_age = 0.5))

b <- sim_abundance() %>%  # pipe approach

  sim_distribution(ays_covar = sim_ays_covar(range = 2000, # diffuse

                                             phi_year = 0.2,

                                             phi_age = 0.9))
```

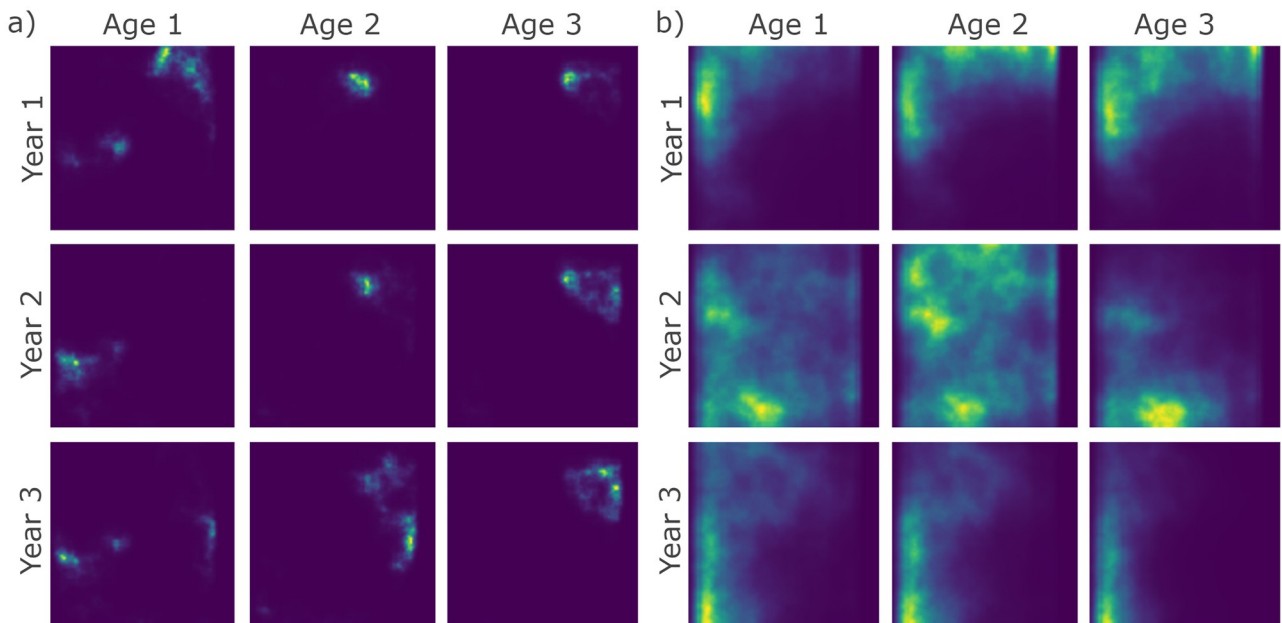

**Fig 3. Distribution plots of simulated populations that form a) tight clusters with stronger correlation through years than ages (default settings), and b) relatively diffuse clusters with stronger correlation through ages than years.** This plot is a facet of plots produced by plot_distribution when supplied simulations from sim_distribution.

The sim_distribution function retains all the data simulated by sim_abundance and adds a data.table [23], named sp_N, with abundance (N) split by age, year and cell. The function also retains the grid object and converts these data into a data.table, named grid_xy, with headers x, y, depth, cell, division and strat. The sp_N object can be merged with the grid_xy data by cell to associate abundance with specific locations, depth, divisions or strata. The plot_distribution function can be used to provide a quick visual of the distribution across ages and years. The code below will generate interactive plots with an Age-Year slider, however, for this paper we present a facet plot of the simulated data (Fig 3).

```
plot_distribution(a, ages = 1:3, years = 1:3, type = "heatmap")

plot_distribution(b, ages = 1:3, years = 1:3, type = "heatmap")
```

## sim_survey

The function sim_survey can be used to simulate data from one survey over a population created using sim_distribution. A default function call is described in Table 4. The sim_survey function simulates the sampling process of the survey and, as such, requires a closure for defining catchability as a function of age and survey protocol settings. Specifically, the q argument requires a closure, such as sim_logistic, for defining the probability of catching specific age groups, trawl dimensions are defined in the trawl_dim argument, and set, length and age sampling effort are defined using the set_den, lengths_cap and ages_cap arguments, respectively. Also note that the min_sets argument imposes a minimum of number of sets to conduct per strata, regardless of its allocation given strata area and set density. This argument imposes a useful constraint for generating data to be analyzed using design-based approaches that require more than one value for the calculation of a mean.

Like sim_abundance and sim_distribution, custom closures can be supplied to sim_survey to impose alternate parametric curves for catchability at age (i.e. a closure including an equation for a dome can be constructed and used in lieu of sim_logistic to impose a dome-shaped catchability). Multiple simulations of the same survey can be run using the n_sims argument, however, requesting large numbers of simulations can be computationally demanding depending on the processing capacity available. Below we use sim_survey to simulate two surveys over a default population, of which one is set-up to have higher set density (set_den) than the other.

```
set.seed(438)

pop <- sim_abundance() %>%

  sim_distribution()

a <- pop %>%

  sim_survey(n_sims = 5,

             set_den = 1 / 1000,

             lengths_cap = 100,

             ages_cap = 5)

b <- pop %>%

  sim_survey(n_sims = 5,

             set_den = 5 / 1000,

             lengths_cap = 500,

             ages_cap = 25)
```

Again, this function retains all the objects listed in the output of sim_distribution and adds data.tables that detail the set locations (setdet) and sampling details (samp). Catchability corrected abundance matrices (abundance at age matrix multiplied by survey catchability), named I and I_at_length, are also produced and added to the output; these matrices are useful for comparing the true abundance available to the survey to abundance estimates obtained using design-based or model-based analyses of the simulated survey data. Specific surveys can

**Table 4. Default sim_survey function call, with descriptions and associated parameter symbols of key arguments.**

| Function call | Description | Symbol |
|---|---|---|
| sim_survey( | | |
| sim, | Simulated spatial population from sim_distribution | |
| n_sims = 1 | Number of times to repeat the survey | |
| q = sim_logistic(k = 2, | Steepness of logistic curve of catchability | $k$ |
| x0 = 3), | Midpoint of logistic curve of catchability (age) | $x_0$ |
| trawl_dim = c(1.5, 0.02), | Trawl dimensions (distance towed, trawl width; km)—i.e. area trawled | $A_{trawl}$ |
| min_sets = 2 | Minimum number of sets to conduct per strata | |
| set_den = 2/1000, | Set density ($km^{-2}$) | $D_{sets}$ |
| lengths_cap = 500, | Maximum number of lengths to collect / set | $M_{lengths}$ |
| ages_cap = 10, | Maximum number of ages to sample / length group / spatial group [1] | $M_{ages}$ |
| age_sampling = "stratified", | Controls whether age sampling is length "stratified" or "random" | |
| age_length_group = 1, | Length group bin size for stratified age sampling (cm) | $l_{group}$ |
| age_space_group = "division") | Spatial scale of stratified age sampling ("division", "strat", "set") | $s_{group}$ |

[1] Length group and spatial group are defined using the age_length_group and age_space_group arguments, respectively. These arguments are ignored if age_sampling is set to "random" and the value supplied to ages_cap represents the maximum number of ages to sample per set.

be explored using the plot_survey function, which uses plotly [25] and crosstalk [30] in the background to link the bubble plot of aggregate set catch to the histogram of lengths and ages sampled to facilitate explorations of set-specific catches (Fig 4).

```
plot_survey(a, which_sim = 1, which_year = 20)

plot_survey(b, which_sim = 1, which_year = 20)
```

As noted above, available RAM may limit the utility of the sim_survey function for running thousands of simulations of the same survey. The sim_survey_parallel function was therefore constructed to facilitate this process. This function is set-up to run multiple sim_survey calls in parallel using the **doParallel** package [31] and, as such, multiple loops can be run using the n_loops argument and, within each loop, multiple simulations can be run (controlled using the n_sims argument). Total simulations will be the product of n_loops and n_sims arguments. If more than one core (cores) is specified, then the simulations will be run in parallel to speed up the process. Low numbers of n_sims and high numbers of n_loops will be easier on RAM, but may be slower. The optimum ratio of n_sims to n_loops will depend on the amount of RAM and number of cores in a given computer. In any case, this function simplifies the process of running thousands of simulations of the same survey and the simulated data can then be supplied to survey-based or model-based analyses that require simulation testing.

## run_strat

Stratified estimates of abundance are obtained by supplying the output from sim_survey to the run_strat function. There are only four arguments in the run_strat function: length_group, alk_scale, strat_data_fun, and strat_means_fun. The length_group argument defines the size of the length frequency bins for both abundance at length calculations and age-length key construction; this argument is set to "inherit", by default, to utilize the length_group value defined

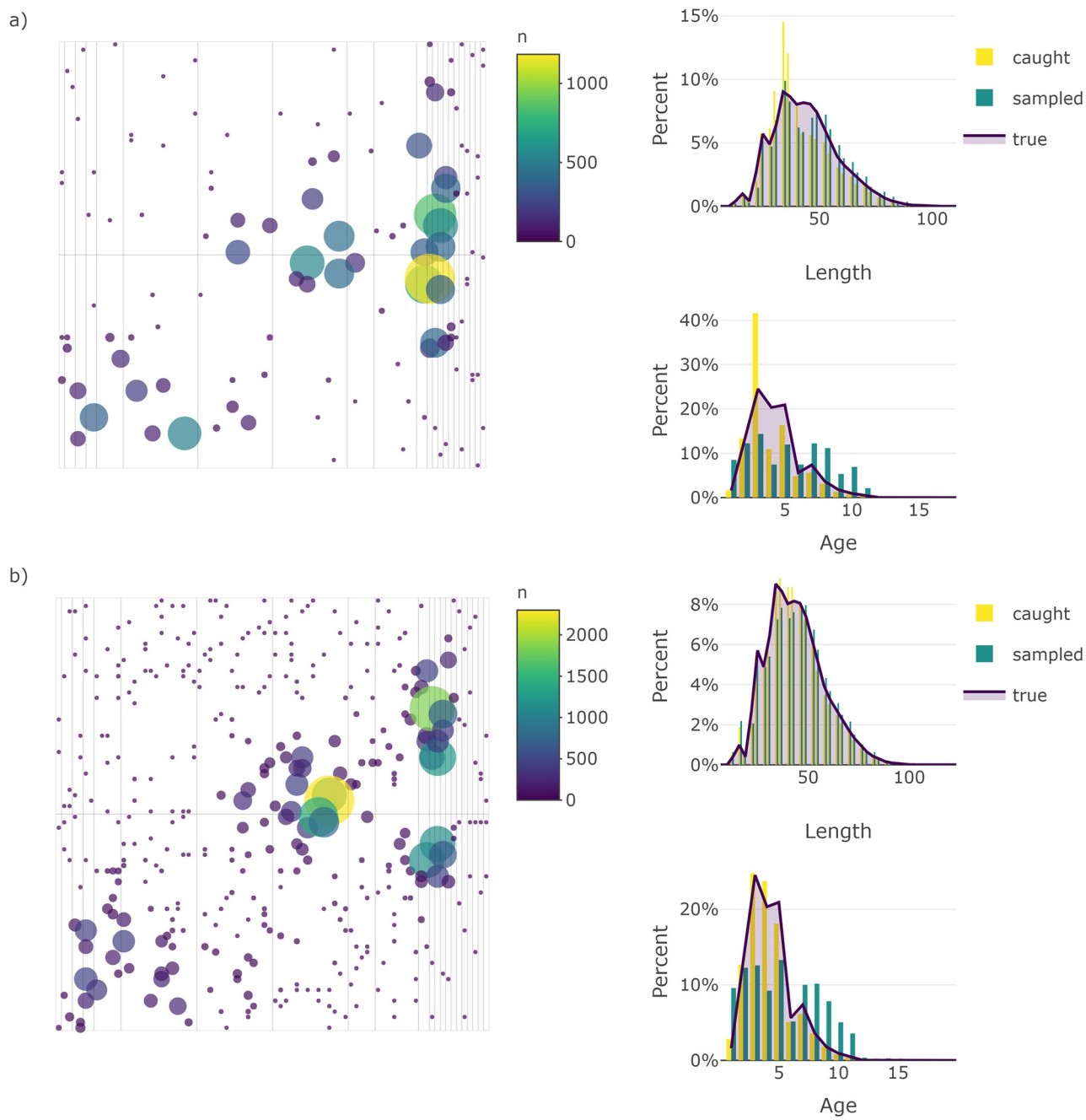

**Fig 4. Bubble plots of abundance and histograms of set catches from a simulated stratified-random survey of a default population under relatively a) low and b) high sampling effort.** Point size and color are scaled by abundance in the bubble plots. Histograms of length and age composition include the distribution of all fish caught next to those sampled overlaid with a line of the true distribution of lengths and ages available to the survey. Note that the first simulation of the survey in year 20 is depicted here. These plots are produced by plot_survey when supplied survey data simulated using sim_survey.

inside the closure supplied to the growth argument in sim_abundance. The alk_scale argument defines the scale at which to construct and apply age-length keys ("division" [default], "strat" or "set"). Finally, strat_data_fun and strat_means_fun allow users to supply custom functions for processing the simulated data and calculating stratified means, respectively; the built in

functions strat_data and strat_means are supplied by default. RMSE of the stratified estimates from run_strat can then be calculated using the strat_error function. Results and error of a stratified analysis of one survey over a population are obtained using the following code (using default values):

```
set.seed(438)

sim <- sim_abundance() %>%

  sim_distribution() %>%

  sim_survey() %>%

  run_strat() %>%

  strat_error()
```

The returned object will include all the objects accumulated through the sim_abundance to strat_error. The run_strat function adds three data.tables called total_strat, length_strat and age_strat that include stratified estimates of total abundance, abundance at length, and abundance at age, respectively. To this, strat_error adds data.tables ending with _strat_error or _strat_error_stats. The _strat_error objects simply contain stratified estimates of abundance (column named I_hat) with corresponding true values of abundance available to the survey (column named I) and the strat_error_stats data.frame includes metrics of mean error (ME), mean absolute error (MAE), mean-squared error (MSE), and root-mean-squared error (RMSE).

## test_surveys

Assuming a stratified analysis as the default method for obtaining an index of abundance, a series of survey protocols can be tested using the test_surveys function. Provided a simulated population from sim_distribution and a series of survey protocols from expand_surveys, this function will simulate and analyze data from each survey using the sim_survey, run_strat, and strat_error functions. Like sim_survey_parallel, this function operates in parallel and allows the specification of n_sims and n_loops, and the product of these two arguments equals the number of times each survey is simulated. Keep in mind that low numbers of n_sims and high numbers of n_loops will be less demanding on RAM, but may be slower, especially if the work is spread across few cores. Because most of the default settings of the functions match the case study settings, the code below will replicate the results from our case study (see S1 Appendix for more detail). The expand_surveys function sets up a series of 175 surveys to test (i.e. all

possible combinations of the set_den, lengths_cap and ages_cap vectors) and the test_surveys function will run 1000 simulations of each survey and compare stratified estimates of abundance to the true abundance available to the survey. While test_surveys is set-up for testing key sampling effort settings (set_den, lengths_cap, and ages_cap), other options can be assessed using independent calls of test_surveys as it accepts several arguments from sim_survey (q, trawl_dim, min_sets, age_sampling, age_length_group, and age_space_group) and run_strat (length_group and alk_scale).

```
set.seed(438)

pop <- sim_abundance() %>%

  sim_distribution()

surveys <- expand_surveys(set_den = c(0.0005, 0.001, 0.002, 0.005, 0.01),

                          lengths_cap = c(5, 10, 20, 50, 100, 500, 1000),

                          ages_cap = c(2, 5, 10, 20, 50))

tests <- test_surveys(pop, surveys = surveys,

                      n_sims = 5, n_loops = 200, cores = 3)
```

Processing time will be system (i.e. amount of RAM and number of cores) and setting (i.e. n_loops and n_sims ratio) dependent. The test_survey function will print a progress bar, generated using the **progress** package [32], which details percent completion and will also include an estimate time of arrival (eta) after the first step of the loop completes. The test_surveys function therefore includes an option for exporting intermediate results to a local directory, via the export_dir argument, and the resume_test function can be used to resume a test_surveys run that had to be stopped part way through the process. The final object produced will be a list that includes all objects from sim_abundance and sim_distribution with the table of survey designs tested (named surveys) and tables produced by strat_error that end with the names _strat_error and _strat_error_stats. These tables include a survey column to allow merging of the survey protocol table with the error tables. Objects produced by sim_survey (set and sampling details) and run_strat (full stratified analysis results) are not retained to minimize object size.

Like other core functions, some convenience functions are included in **SimSurvey** for creating interactive plots of the results from test_surveys. For instance a series of plotting functions ending in _fan produces fan charts where stratified estimates of abundance from each simulated survey are converted into a series of quantiles to depict the probability that estimates fall within a particular range. True values of abundance available to the survey are overlaid on the series of probability envelopes. These plots help visually assess the level of precision and

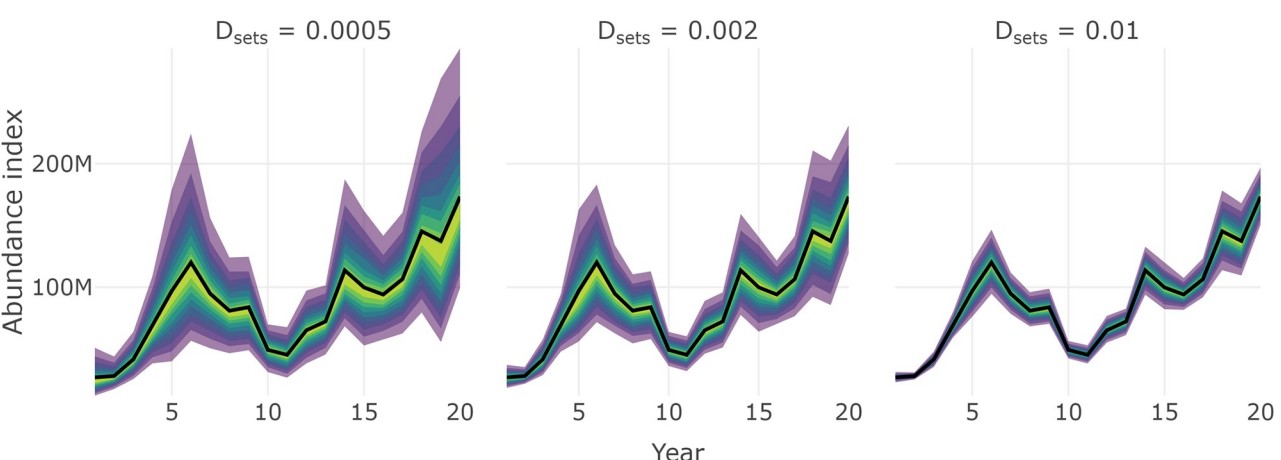

**Fig 5. Fan chart of stratified estimates of the trend in total abundance from surveys with different set densities, $D_{sets}$.** The thick black line indicates the true trend in the total population available to the survey and the yellow to purple color gradient represents a range of probability envelopes from 10% to 90%. This plot is a facet of plots produced by plot_total_strat_fan when supplied results from test_surveys.

bias from a specific set of survey protocol; ideally, the probability envelopes will be tightly centered around the true values. The three lines of code below will produce interactive fan charts for stratified estimates of total abundance, abundance at length and abundance at age, respectively (e.g. Figs 5, 6 and 7).

```
plot_total_strat_fan(tests)

plot_length_strat_fan(tests, years = 1:20, lengths = 1:100)

plot_age_strat_fan(tests, years = 1:20, ages = 1:10)
```

The relative performance of the surveys tested can be compared using plot_survey_rank and plot_error_surface. The plot_survey_rank function produces a divergent dot plot of the results which ranks the surveys by RMSE, where lower values indicate sampling strategies that minimize bias and maximize precision. Using the which_strat argument, the plot can be focused on total, length or age based stratified results (Fig 8). The plot_error_surface displays the age based stratified results by plotting a surface of RMSE (z-axis) by set (drop down selection), length (y-axis), and age (z-axis) sampling effort. The sampling effort axes can be rule or sample size based (plot_by = "rule" or plot_by = "samples", respectively; Fig 9).

```
plot_survey_rank(tests, which_strat = "length")

plot_error_surface(tests, plot_by = "rule")
```

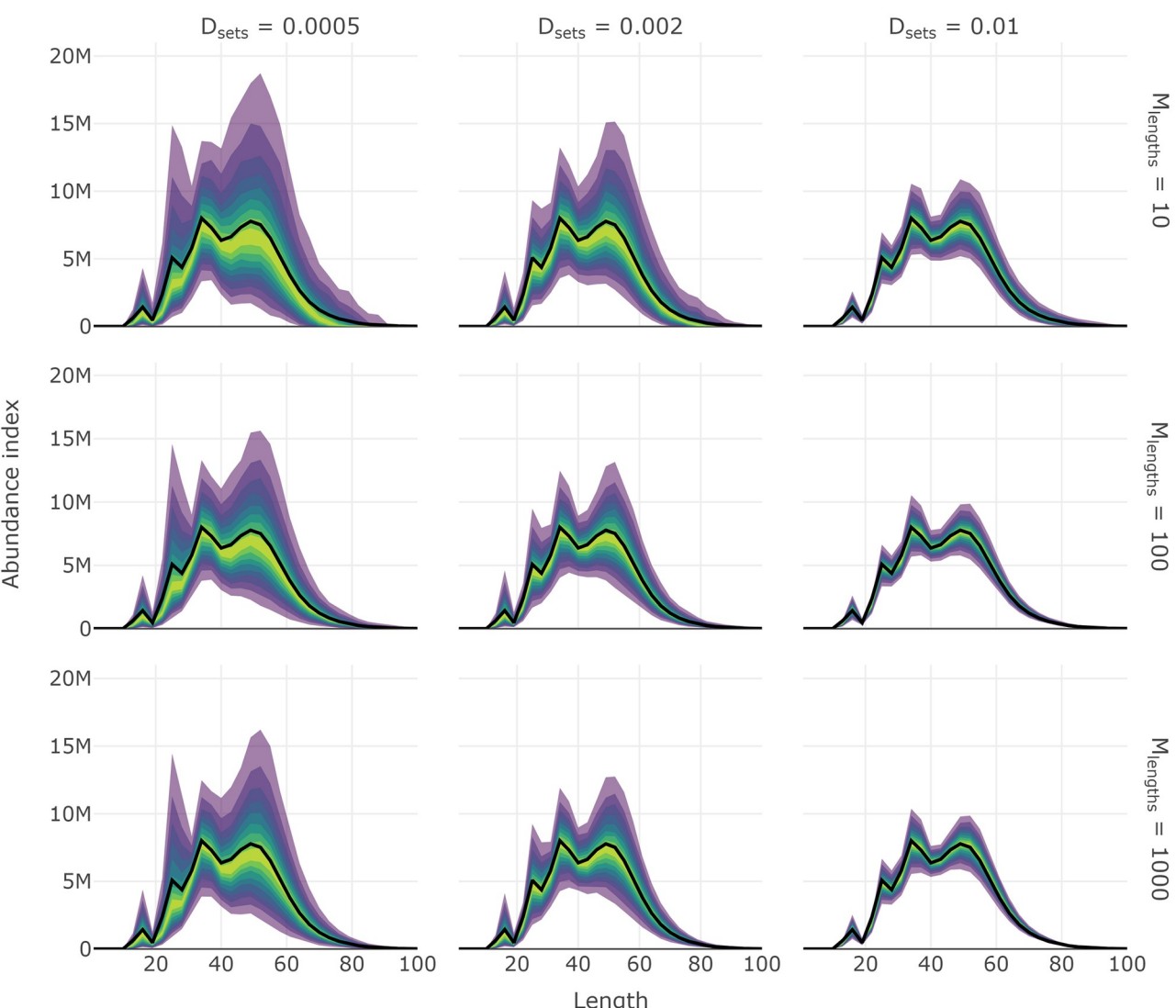

**Fig 6. Fan chart of stratified estimates of abundance at length from year seven of the simulation from surveys with different set densities, $D_{sets}$, and the maximum number of length samples, $M_{lengths}$.** The thick black line indicates the true trend in the total population available to the survey and the yellow to purple color gradient represents a range of probability envelopes from 10% to 90%. This plot is a facet of plots produced by plot_total_strat_fan when supplied results from test_surveys.

## Interpretation

Results shown in the **Core functions** section demonstrates one use of the **SimSurvey** package that is focused on evaluating the efficacy of increasing the sampling effort of a stratified random survey of a cod population (see S1 Appendix for details). These results largely align with expectations from sampling theory that 1) design-based estimators for stratified random surveys are unbiased, and 2) precision is increased by increasing the number of primary sampling units [33]. Specifically, estimates of total abundance and abundance at length are centered around true values and their probability envelopes tighten as set density increases (Figs 5 and 6). Our case study results also echo the growing body of literature which concludes that extra sub-sampling is an ineffective means of improving estimates relative to sampling more

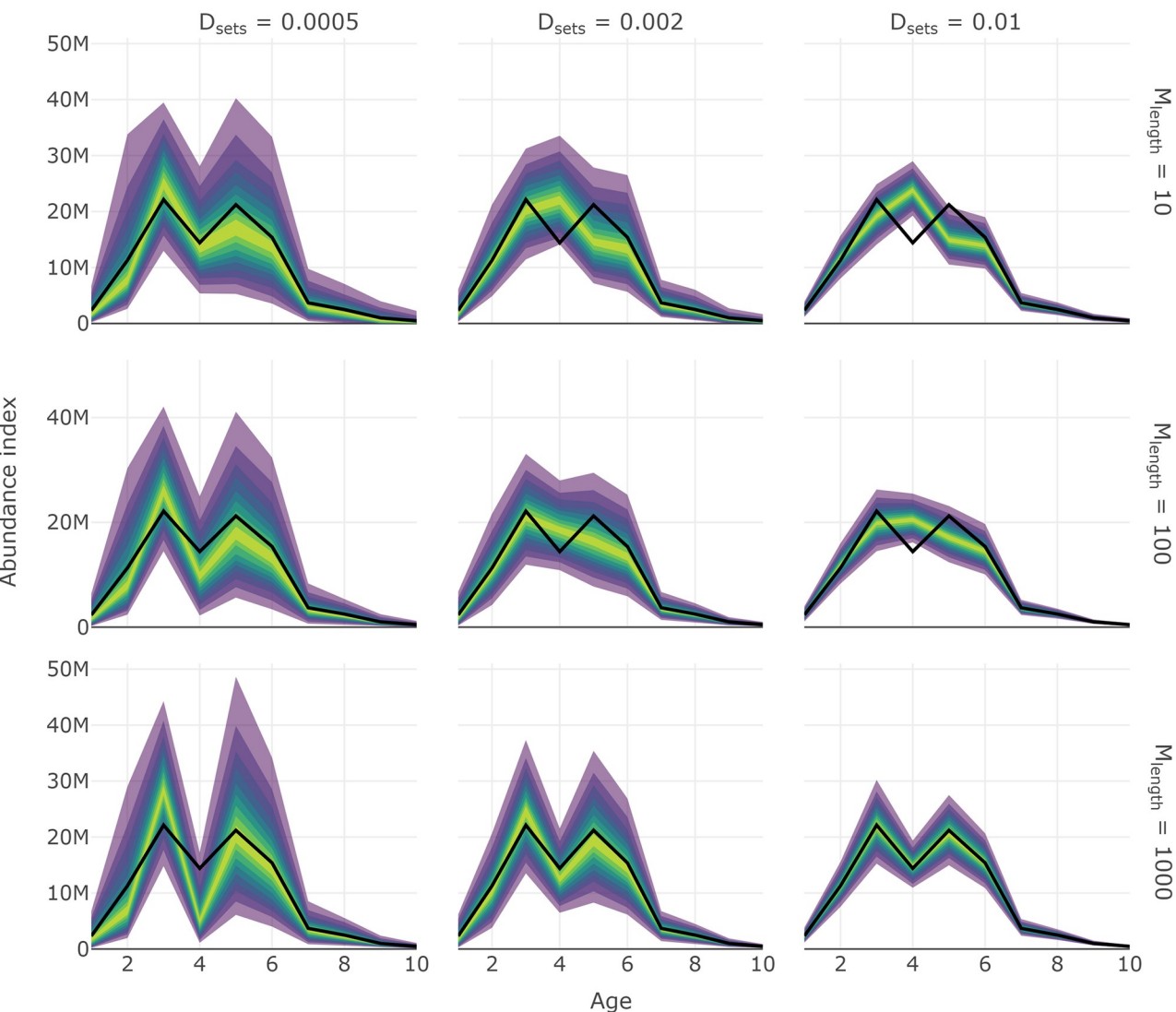

**Fig 7. Fan chart of stratified estimates of abundance at age from year seven of the simulation from surveys with different set densities, $D_{sets}$, and the maximum number of length samples, $M_{lengths}$.** Number of ages sampled per length group, $M_{ages}$, was 10 in all scenarios. The thick black line indicates the true trend in the total population available to the survey and the yellow to purple color gradient represents a range of probability envelopes from 10% to 90%. This plot is a facet of plots produced by plot_total_strat_fan when supplied results from test_surveys.

locations [10–12,34–36]. This is exemplified by the relatively large drops in RMSE when set density is increased compared to when sub-sampling effort is increased (Figs 8 and 9). Moreover, it appears that it is more advantageous to measure fewer total fish at more locations than measuring many fish at fewer locations (Fig 8). Again, this result was expected because the size-structured spatial clustering imposed by the simulation causes set-to-set variation to be greater than within-set variation in characteristics such as length and age. This structure was imposed because fish caught together tend to have similar characteristics and it is well known that this reduces the effective sample size [10]. Increasing sub-sampling effort, however, should not lead to poorer population estimates like those observed for abundance at age (Figs 7 and 9). Bias, in particular, appears to be a problem with stratified estimates of abundance at age under certain scenarios (Fig 7); in contrast, there appears to be little bias in the stratified

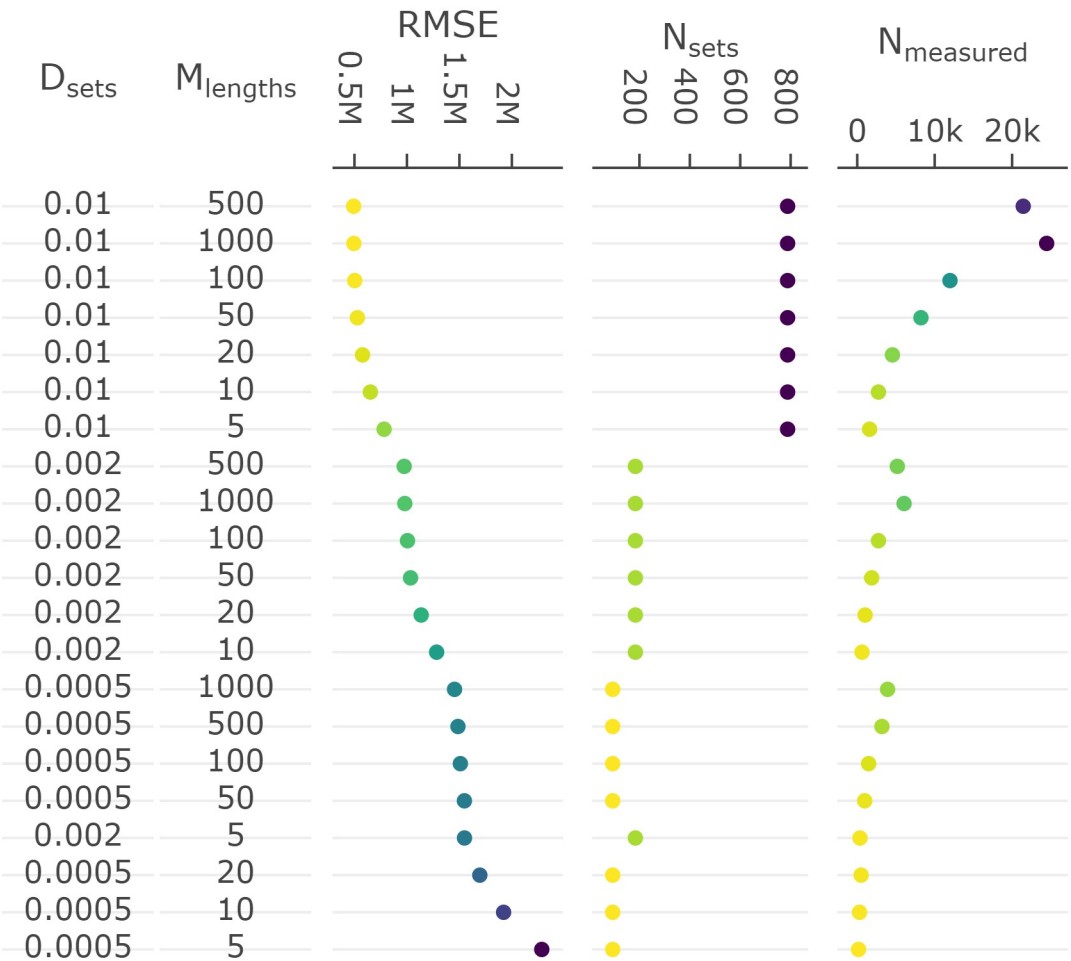

**Fig 8. Divergent dot plot of the precision and accuracy (root-mean-squared error; RMSE) of length based stratified estimates of abundance, and total sampling effort (number of sets [$N_{sets}$] and length measurements [$N_{measured}$]), under various sampling protocols (set density [$D_{sets}$] and maximum number of lengths measured per set [$M_{lengths}$]).** Records are ranked by lowest to highest RMSE score. Within each plot, a yellow to purple color gradient is applied from lowest to highest value as an additional visual aid. Note that the exponent format of the axes defaults to SI unit symbols (e.g. M for million).

estimates of abundance at length (Fig 6). This result was unexpected as these estimates of abundance at age were presumed to be unbiased. The contrast between the length and age based analysis indicates that the problem lies with the intervening age-length-key.

Age-length keys have long been used in conjunction with length frequency tables to estimate the age distribution of fish populations over large scales [37]. By default, sim_survey spreads length-stratified age sampling across the division and run_strat constructs and applies age-length keys at the division scale. This default was chosen because it is standard protocol for the actual survey the case study is based on. There is, however, a potential cost to the spatial scale of the key. Namely, it is unlikely that one age-length key is representative for the whole region as the probability of being a specific age given length varies in space [38]. Because different age groups sometimes occur in different places, the translation of lengths to ages may be biased by the samples used to generate the age-length key. In short, multi-stage sampling of a clustered population violates the assumption that the age-length key is generated from a simple random sample [21]. Bias introduced by this assumption can be resolved by explicitly accounting for the hierarchical sampling by using designed-based estimators [21]. For the case study, a

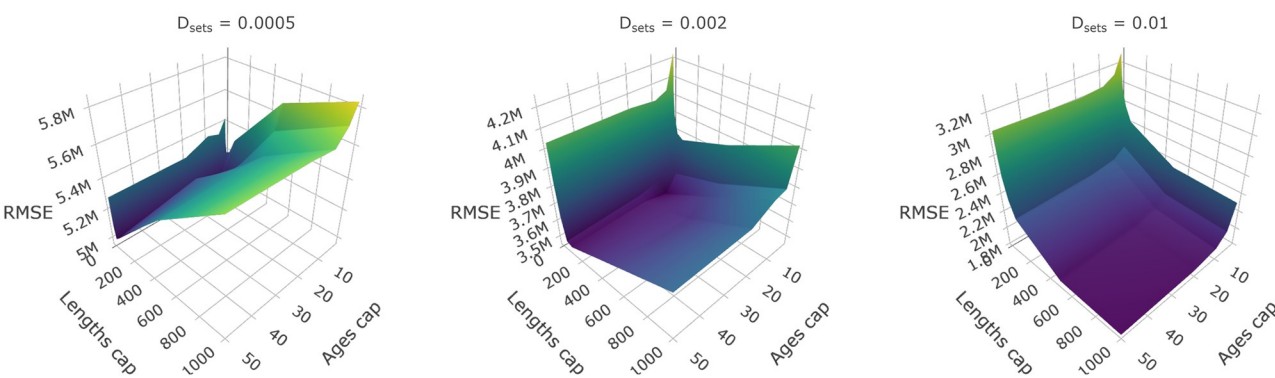

**Fig 9. Surface plots of root-mean-squared error (RMSE) from an array of surveys with different sampling protocol.** Panels represent surveys with different set densities ($D_{sets}$), x-axes represent the maximum sampling effort of lengths per set ($M_{lengths}$), and y-axes represent the maximum number of ages to collect per length group ($M_{ages}$). This plot is a facet of plots produced by plot_error_surface when supplied results from test_surveys. Note that RMSE scales are different across $D_{sets}$ panels. Note that the exponent format of the axes defaults to SI unit symbols (e.g. M for million).

proper design-based estimator will need to account for cluster sampling in a stratified random survey. Following recommendations in Aanes and Vølstad [21], we used **SimSurvey** to 1) conduct a survey with concurrent length and age sampling at every set, 2) construct and apply age-length keys on a set-by-set basis, and 3) weight the resultant age frequencies at each set using stratified random estimators. This test was accomplished using this code:

```
alt_surveys <- expand_surveys(set_den = 2 / 1000,

                              lengths_cap = c(5, 10, 20, 50, 100, 500, 1000),

                              ages_cap = c(1, 2, 3, 5, 10))

alt_tests <- test_surveys(pop, surveys = alt_surveys,

                          n_sims = 5, n_loops = 200, cores = 3,

                          age_length_group = 3,

                          age_space_group = "set",

                          alk_scale = "set")
```

This is a minor modification of the code used in the **test_surveys** section whereby the same simulated population was used (pop object) but set density scenarios were reduced to one option for simplicity, and age sampling protocol and stratified analysis options were changed. That is, 1) age_length_group and age_space_group were changed to simulate a survey that

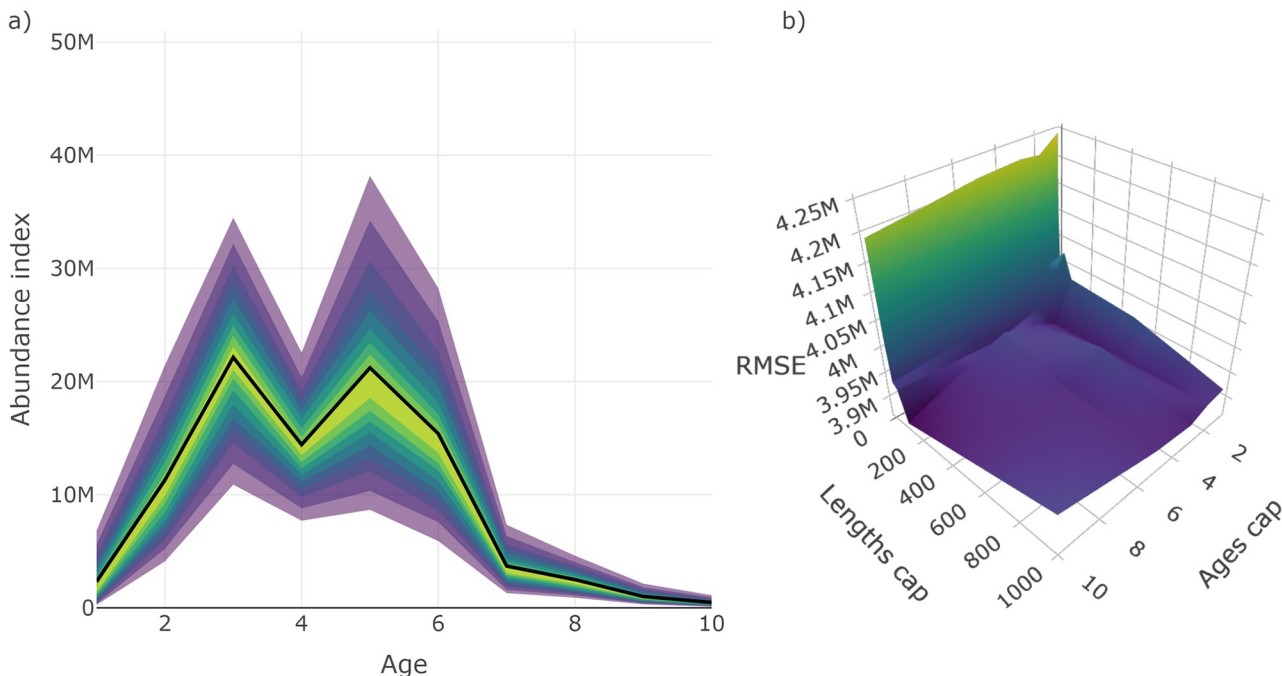

**Fig 10. Main results from an alternative design-based sampling and analysis procedure that accounts for multi-stage cluster sampling of ages.**
Specifically, a) is a fan chart of stratified estimates of abundance at age from year seven of the simulation from a survey with a set density ($D_{sets}$) of 0.002 sets / km$^2$, maximum number of length samples ($M_{lengths}$) of 100, and maximum number of age samples per 3 cm length group per set ($M_{ages}$) (the thick black line indicates the true trend in the total population available to the survey and the yellow to purple color gradient represents a range of probability envelopes from 10% to 90%); and b) is a surface plot of root-mean-squared error (RMSE) from an array of surveys with set density ($D_{sets}$) fixed to 0.002 sets / km$^2$ but different sub-sampling effort, where the x-axes represent the maximum number of lengths to measure per set ($M_{lengths}$), and y-axes represent the maximum number of ages to collect per 3 cm length group per set ($M_{ages}$).

conducts concurrent length and age sampling at every set (e.g. collect age samples from a maximum of 1 fish per 3 cm length group per set), 2) alk_scale was changed to "set" to construct and apply age-length keys on a set-by-set basis, and 3) the resultant age frequencies at each set are weighted accordingly using the design-based estimators built into run_strat. Again, results are visualized using plot_age_strat_fan and plot_error_surface:

```
plot_age_strat_fan(alt_tests, years = 1:20, ages = 1:10)

plot_error_surface(alt_tests, plot_by = "rule")
```

Unlike the default approach to collecting and analyzing age samples, results from this test appear to be unbiased and additional sub-sampling effort beyond a certain point appears to have little to no effect on the estimates (Fig 10). This is an important result with real-world implications as they imply that 1) inference from the actual survey can be improved by altering the sampling and analysis of ages and, 2) valuable survey time can be saved by cutting back on sub-sampling effort. This example demonstrates how **SimSurvey** can be used to reveal problems and explore solutions to the sampling designs of fisheries-independent surveys.

## Discussion

Though somewhat narrow in scope, results from our case study demonstrate why it is important to evaluate the design and analysis of complex surveys. Specifically, we tested the design and analysis of a multi-staged stratified random survey of a cod population. This type of sampling is common in fisheries research, however, the clustered nature of the samples are rarely taken into consideration [9]. Instead, it is commonly assumed that individuals are randomly drawn from the target populations even though such sampling is virtually impossible to accomplish in practice. The assumption of simple random sampling implies that the characteristics of fish caught at the same location are independent and, if estimators that assume independence are used, this can introduce bias and underestimate variance [9,21]. Moreover, this assumption may contribute to the perception that more sub-samples will lead to better population estimates. Increasing sub-sampling effort, however, has been shown to be an ineffective means of improving estimates because samples at the same location are usually correlated; the best way to increase the number of independent samples is to sample more locations [10–12,34–36]. Results from the case study reiterate these points and indicate that there is room for improvements to the actual survey the case study was based on. Specifically, it appears that valuable human resources may be wasted by collecting excessive samples of correlated metrics, and abundance at age estimates may be biased by the assumption that age samples are from a simple random sample. We have used **SimSurvey** to identify solutions to these problems and, given current results, we suggest that time can be saved by decreasing sub-sampling effort and bias can be reduced by using an approach that account for the hierarchical design of the survey.

### Research opportunities

The case study used here provides one example of how **SimSurvey** can be used to simulation test the design of fisheries-independent trawl surveys. Default settings can, of course, be modified to emulate data from an array of different surveys of different populations (see S2 Appendix for guidance on how to modify default settings to suit specific needs). The package therefore facilitates the exploration of a range of research questions. Below we outline some examples where the **SimSurvey** package may aid future research efforts.

**Design or model-based approach.** The analysis of data from fisheries-independent surveys have generally been confined to design-based mean and variance estimates of abundance using standard formulae (e.g. [33]). Nevertheless, there has long been interest in using model-based approaches to improve abundance estimates (e.g. [19,39,40]). **SimSurvey** can serve as a convenient tool for simulation testing mean and variance estimates provided by a range of different approaches (design-based analyses, bootstrap estimates, generalized additive models, geostatistical models, etc.). Moreover, the full analytical pathway for obtaining age-disaggregated estimates of abundance has rarely been simulation tested. Existing and future approaches for calculating age-based indices of abundance can be simulation tested using **SimSurvey**.

**Growth analyses.** Assessing ages for a large number of fish is very time-consuming and, as such, length-stratified sampling is often used to estimate age frequencies of fish populations. The resultant sub-sample is used to construct an age-length key (i.e. the probability a fish is a specific age given length) and age frequencies are obtained by applying this key to length-frequencies obtained via more expansive random sampling. One age-length key is typically assumed to be representative of the whole stock area, however, spatial variability in the relationship may introduce bias in abundance-at-age estimates [21,38]. Results from the case study (see S1 Appendix) reiterate this point and **SimSurvey** may serve as a platform for testing potential model-based solutions to this problem (e.g. [38]).

**Random or stratified sampling.** **SimSurvey** can be used to compare the precision and bias of population estimates obtained using random or stratified sampling. Simple random sampling can be implemented using a grid with one strata (e.g. make_grid(depth_range = c(0, 1000), strat_breaks = c(0, 1000), strat_split = 0)). Sub-sampling of ages can also be random rather than length-stratified by setting the age_sampling argument in the sim_survey function to "random" rather than "stratified". This can facilitate research similar to work presented in Puerta et al. [15].

## Future directions

Up to now, the package has focused on the effects of sampling design on the precision and bias of population estimates obtained from fisheries-independent surveys; however, the costs associated with sampling has yet to be considered. In future iterations of **SimSurvey**, we hope to add options for integrating data on the time and monetary costs associated with each level of sampling (sets, length measurements, age determination) to facilitate cost-benefit analyses. We also realize that a single fisheries-independent survey may have multiple goals as data obtained are often used to assess multiple species or to conduct community analyses. We will therefore endeavor to add functions for simulating multi-species surveys. Another limitation of the package is that we have yet to implement alternatives to random or stratified random survey designs (e.g. systematic sampling); expanding these options would allow for a more comprehensive evaluations of various designs. Finally, it would be useful to add an option for testing the consequences of surveys with partial coverage of a population as survey coverage is a frequent concern in stock assessment.

## Assumptions

Like any model, this simulation is a simplification of a much more complex reality. For instance, the population is assumed to aggregate by age-class and be uniformly distributed within a cell, instead fish may aggregate by length and form finer-scale clusters. The survey is also an instantaneous snapshot of the population, meaning that the population is assumed to be in the same location from the beginning to the end of the survey. Also, fish are aged at random within length bins and ages are estimated without error. Finally, area trawled is assumed to be perfectly standard. These assumptions, plus a range of others, will surely under-represent the natural variability of fish populations and survey protocol. Nevertheless, the **SimSurvey** package provides a relatively complex and flexible operating model for simulating survey data from a population that varies across age, year and space dimensions.

## Summary

The **SimSurvey** package serves as a tool for simulating sample surveys of dynamic populations that vary across ages, time and space. The core of the simulation is based on the widely used cohort equation and, even though the processes that define recruitment and total mortality are simple, a wide range of stock dynamics can be simulated by changing a few parameters. This base population can then be distributed through a grid and relationships with depth and correlation across ages, years and space can be defined. Together, two functions (sim_abundance and sim_distribution) are capable of simulating a wide range of populations with different life histories, depth associations and spatial properties. The next step to generating data similar to actual observations is to conduct a survey. In this package we implement a function, sim_survey, that conducts a survey of the population. The sampling process is governed by the area covered by the trawl as well as age-specific catchability. Sub-sampling protocol (length and age sampling) can also be varied. As such, data from a wide range of surveys can be simulated.

A large number of statistical models can be tested using these simulated data, but, implementing a variety of analytical approaches was outside the scope of this package. Instead we focus on analyzing simulated stratified-random survey data using a design-based stratified analysis. A stratified analysis is facilitated using the run_strat function and the precision and accuracy of the results (e.g. RMSE) can be calculated using strat_error. This is a simple and widely-used analysis, and the speed at which it runs allows for a wide range of survey designs to be tested, via the test_surveys function, in a reasonable time-frame.

Simulation testing is an important tool in the field of fisheries science as the inferred status of fish stocks hinge on the data and models used to assess fish populations. Simulations provide an opportunity to explore survey and model performance, and such explorations are becoming increasingly important as model complexity increases. It is also important to continually assess the efficacy and efficiency of sampling programs given their costs and the constant scrutiny of the value added by such surveys. These are some of the reasons multiple simulation frameworks, including **SimSurvey**, have been developed to test the design and analyses of complex surveys. We have made **SimSurvey** as open and accessible as possible to allow the broader community to validate, reuse and improve this package. We hope that open-source sharing will extend the value of such simulation frameworks and we encourage users to extend the package for their own needs and contribute to future versions.

## Supporting information

**S1 Appendix. Case study.**
(DOCX)

**S2 Appendix. Parameterisation.**
(DOCX)

**S3 Appendix. Age-year-space covariance.**
(DOCX)

**S4 Appendix. Stratified analysis equations.**
(DOCX)

**S5 Appendix.**
(HTML)

**S1 Fig.**
(PNG)

## Acknowledgments

This work has benefited from valuable feedback from numerous colleagues, including Aaron Adamack, Alejandro Buren, Noel Cadigan, Karen Dwyer, Geoff Evans, Paul Higdon, Danny Ings, Mariano Koen-Alonso, Joanne Morgan, Derek Osborne, Pierre Pepin, Dwayne Pittman, Don Power, Craig Purchase, Martha Robertson, Mark Simpson, Brad Squires, Don Stansbury and Peter Upward. We also thank Dave Cote and Joanne Morgan for providing constructive comments on a previous version of this manuscript. Finally, the package and the manuscript were greatly improved by feedback from editor Daniel Duplisea and two anonymous reviewers. This work was supported by the NSERC visiting-fellow program and Fisheries and Oceans Canada.

## Author Contributions

**Conceptualization:** Paul M. Regular, Brian Healey, Fran Mowbray.

**Formal analysis:** Paul M. Regular.

**Funding acquisition:** Paul M. Regular, Fran Mowbray.

**Methodology:** Paul M. Regular.

**Project administration:** Fran Mowbray.

**Software:** Paul M. Regular, Gregory J. Robertson, Jonathan Babyn.

**Supervision:** Gregory J. Robertson, Keith P. Lewis, Brian Healey, Fran Mowbray.

**Validation:** Paul M. Regular.

**Visualization:** Paul M. Regular.

**Writing – original draft:** Paul M. Regular.

**Writing – review & editing:** Paul M. Regular, Gregory J. Robertson, Keith P. Lewis, Brian Healey.

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
