## [Decision Letter · Decision Letter 0]

13 Nov 2019

PONE-D-19-26904

SimSurvey: an R package to optimize the design and analysis of fisheries surveys by simulating spatially-correlated fish stocks

PLOS ONE

Dear Dr. Regular,

Thank you for submitting your manuscript to PLOS ONE. After careful consideration, we feel that it has merit but does not fully meet PLOS ONE’s publication criteria as it currently stands. Therefore, we invite you to submit a revised version of the manuscript that addresses the points raised during the review process.

These two thorough reviews raise excellent points that should greatly improve the utility of the package if they can be addressed. Please consider each point particularly keeping in mind the research content of the manuscript beyond a description of the software.

We would appreciate receiving your revised manuscript by Dec 28 2019 11:59PM. To enhance the reproducibility of your results, we recommend that if applicable you deposit your laboratory protocols in protocols.io, where a protocol can be assigned its own identifier (DOI) such that it can be cited independently in the future. For instructions see: http://journals.plos.org/plosone/s/submission-guidelines#loc-laboratory-protocols

We look forward to receiving your revised manuscript.

Kind regards,

Daniel E. Duplisea, PhD

Academic Editor

PLOS ONE

Journal Requirements:

Reviewers' comments:

Reviewer's Responses to Questions

**Comments to the Author**

1. Is the manuscript technically sound, and do the data support the conclusions?

Reviewer #1: Yes

Reviewer #2: No

2. Has the statistical analysis been performed appropriately and rigorously? 

Reviewer #1: Yes

Reviewer #2: N/A

3. Have the authors made all data underlying the findings in their manuscript fully available?

Reviewer #1: Yes

Reviewer #2: Yes

4. Is the manuscript presented in an intelligible fashion and written in standard English?

Reviewer #1: Yes

Reviewer #2: Yes

5. Review Comments to the Author

Reviewer #1: This paper describes the R package “SimSurvey” which is a package to test the design and analysis of fisheries independent surveys.

I found the paper very interesting and the package of great potential value and interest to many fishery scientists. However, I have several comments that I would like the authors to address first before I recommend acceptance of this paper. My main comments relate to:

1. Verify some of the equations and model description used in the model. I think that some of them lack description to fully understand what was done and few others are misleading, even incorrect. The detail can be found below but in general, they are linked to the problem of bias correction for log-normal distribution, re-scaling”, moving from abundance at age to abundance at length.

2. Add additional explanation to justify the choices. E.g. why population dynamics is modeled independently of the spatial distribution. Is this reasonable, limitation, etc.

3. Few suggestions (add-ons, renaming) to increase the generality of the paper (without requiring too much work)

Detailed comments:

Abstract:

What about the flexibility to include new sampling strategies (user defined one) or including new estimation approaches

Introduction:

L43: quality of

It would be good to add few references here e.g. L50,

Method:

L69-73: I think it would be good to add the reference to the specific function name listed in Table2. This way, I would be more explicit and avoid possible confusion. For example, in my case, when I first read the section header “simulate abundance”, I was expecting to see the way you dealt with the spatial distribution of abundance.

L94&95&104: I just want to make sure that the bias correction has been applied to these two equations. If logN (or logZ) is normally distribution with mean mu and sigma, then the N (or Z) is lognormally distributed with mean exp(mu+sigma^2/2). Therefore, when you write “mean” in table 3 for R and Z, I hope that you modified in your actual equation for log(Z) to:

log(Z) ~ normal(log(mean)-sigma^2/2, sigma)

In other word, the mean(R) is not simply exp(meanR_log_scale). exp(meanR_log_scale) is the median of the distribution of R.

L107: there is not enough description to understand how you converted the abundance at age to abundance at length. What were the choice of length bin groups? Every 5 cm, 10cm? Such information is also missing in the tables. Please add that. But then you have to calculate some cumulative distribution (until length l) for the length at age distribution (using the VB growth equation) and do some subtractions to be able to allocate the abundance at age to each length group.

See equation A.1.14 in here for example. https://www.nwfsc.noaa.gov/news/events/program_reviews/documents/C.2_Methot_Wetzel_SSTechnicalDescription.pdf

“Simulate spatial distribution”:

I have a few comments on this section:

1. First of all, I think it is important to state that you assume that the population dynamic model and the spatial distribution is independent i.e. you do not have a spatially-explicit population dynamics model. And talk about what it means (is it realistic, limitation, etc).

2. L161. I think it would be better to rename “depth” as “main covariate influencing the species distribution”. And “depth” happens to be one of them for many species, but there are cases when this is not the case.

3. L161. I think it would be good to say upfront that there are two ways of defining the spatial structure. Using the function in the package or user defined

4. L161: I think another possibility is to generate a random field by using the package “Randomfields” for example. This way you can generate a map where “depth” is patchily distributed (maybe more like an island type case study)

5. L163-164: please add more description about the division or strata. How can we set it up and what do you specifically mean by division and strata? Is one nested within the other, or not necessarily? Your examples are based on Atlantic Ocean and people in other regions might not be familiar with how these divisions are created.

6. L163: Why only focus on “depth-based strata”? I think it would be good to allow the user to choose their own stratification approach. It could be depth based as you did (which happens most often in surveys) but it could technically be any other thing (user supplied). This allows more flexibility.

7. L167: the equation is misleading and I am not sure it is right based on what is written in the text. You mentioned later on L178: that you “re-scaled” so that the total number of fish in a specific year and age across space is equal to the single number from the population dynamics model. If so, the re-scaling should be done in the identify (natural) scale, not in log scale. In log scale, even if ,, sums to zero across space, the sum in the identity scale won’t match. This is often refer to as “bias correction” for the log normal models. And you should ideally show how the rescaling was done in terms of the equations too.

8. L167: this “depth preference” function is very simplistic and gives only very “smooth” symmetric distribution. More often, fish have a skewed depth preference: often right-skewed.

9. L173-174: I think it might be worth adding, in simpler terms, the meaning of the spatial smoothing and scaling parameters.

10. L178: it is another question of scaling. How did you exactly do the scaling? In the identity scale? By dividing my the sum of the effects? I am asking this because depending on how you did the rescaling, your correlation structure in space and age might have been affected and is not the same as the one specified in L172. Did you verify that?

Table 4. “group_ages”. Ok but how is the variance controlled for the other age classes?

L189: “user supplied”. This is a good feature. However, I think it is important to mention here that user have to make sure that they use the correct projection method to ensure that each grid is of the same dimension.

L216: reference to figure 3?

L221: there is not “group_years” argument in Table 4.

L237: “this function”. I think it would be better to replace “this function” with “sim_distribution” as you do not mention the word “sim_distribution” in the sentence above this.

L158-251: In general, I think re-organising this section using sub-section headers could be useful. Just to guide the readers

L254: you say that sampling is stratified random but SRS is also an option based on Table5. Please correct.

L257: what does this mean? Does this control the number of set but how is this calculated?

L257: I do not see how you control for the total number of set in the survey? How do you control it?

L261: I think you should mention here that you can also force the sample size (as seen on table 5). Moreover, in table 5, it would be good to set-up a “ages_min” for the minimum number of ages to sample […]” so that it gives the ability to fix the sample size if needed by writing the same value for “ages_min” and “ages_cap”.

L261: How are you making sure that the number of sampled fish for that specific cell, age and year won’t be above the total number of fish in that cell, age, and year? The probability value could be close to one and if you fish in a few a time, then you are at risk. Especially because your population dynamics model is not spatially explicit and is completely independent of the distribution function itself i.e. you can technically fish out all the fish in an area but it will be populated back the year after the way you implemented in this study… Maybe you need to put a condition (or just a note) for general users to make sure that this probability value is much below 1?

L267: I recommend to clarify something here. 1. Depending on the number of fish caught? What do you mean? What is the rule you used? 2. The way you coded, sample by age is first decided, then the corresponding length is calculated, then age-subsample is determined. In reality, length sample is taken in the field, then age sub-samples are taken. While similar, I do not think it always equal. Especially, when you start including some correlation structure in the sampling. By the way, did you consider including some correlation structure in the sampling process to make more realistic?

L275 Table 1 on should be table 5

L275: Table5: “age_sammpling” should be “age_sampling”

L275: “min_sets” you have not described it yet and what is it? You have sample from all cells? If no, this is not realistic.

L279: Table5 not Table1?

L285-286: Could you be more specific on how custom closures can be supplied and where?

L306: how are these catchability corrected abundance matrices calculated? It is important to write this information somewhere (or write “please refer to the section “Stratified analysis” for further information on the calculation of abundance indices”) or something alike and Appendix S3.

L336: I think it would be good to say that other methods exist and people can use it in this package (maybe)?

L421: color gradient. Even though it is obvious it might be good to say green to purple gradient.

L427: instead of “sampling protocol”, I think it would be more meaning full to say the maximum number of length samples.

L452: say that the color ramps from yellow to purple

S1 appendix: missing figure in S1

Reviewer #2: This manuscript describes an R package called SimSurvey. The package includes a set of functions for simulating point-based fisheries survey designs, e.g. bottom trawl surveys, for estimating abundance indices. It focuses on number of stations and number of fish sampled ignores other constraints such as distance between stations and day-time duration which impose strong constraints on real surveys. The functions included allow the user to first simulate age-and length structured population dynamics, distribute individuals randomly in space (assuming a certain correlation structure), carry out a survey and finally calculate abundance indices from the simulated data. The package will likely be of interest to researchers wanting to explore the precision achievable with different survey designs. My comments regarding the package and the presentation are summarised below.

1. Optimization

The title announces a package for optimizing survey designs. As far as I can see the package does not allow survey design optimization, neither in terms of defining survey strata nor in terms of number of stations per stratum. The strata are defined by the user. The only option available for the number of stations is proportional to stratum surface; the user sets the minimum number of stations taken in the smallest stratum. It would be useful to be able to specify the total number of stations and test different allocation schemes, such as proportional to surface area (implemented), equal number per stratum, Neyman allocation (accounting for surface area and abundance variability), etc.

Please consider revising the title (e.g. “compare” and instead of “optimize)” and spell out the available sampling design options.

2. Manuscript structure

The manuscript might be easier to follow if the manuscript was restructured: 1) Model description, 2) Using SimSurvey. The later section would then group all example code which could again be subdivided into running simulations and exploring results (plot functions).

3. Parameterisation

To use SimSurvey for a real world problem realistic parameter values are needed. The package comes with default values chosen for a particular case study. However, no mention is made in the manuscript how to choose appropriate values for the many model parameters to tailor the simulations to a population of interest. I suggest the authors add a section on parameterisation and a table summarizing all parameters with a column specifying how to parameterize. For example, parameters for population dynamics and growth could probably be taken from the literature (or a stock assessment report). However this is not possible for the parameters of the spatial distribution function sim_distribution() such as correlation between ages etc. Ideally the package would include a fitting function for estimating these parameters from actual data. These input data could come from a pilot survey and include location (lat, long) and numbers by length/age.

Minor issues

- line 93: I assume there is an age plus but this needs to be mentioned. Also, please specify how you set the initial numbers for plus group ().

- line 121: I don’t understand the explanation of a closure. What do you mean by “return functions”? Do you mean it returns an object with different attributes?

- line 125: the number of right and left brackets is unbalanced, please check

- line 126 “This structure was chosen to avoid the repeated specifications of ages and years”. As far as I can see the example code only specifies years, not ages.

- line 227 Please explain what a pipe is and how it is used. In the example I understand that the output of sim_abundance( ) is provided to (piped) into sim_distribution(). I am unclear what the object b contains. Is it the result of sim_distribution()?

- There is no table 1, please revise table numbering.

6. PLOS authors have the option to publish the peer review history of their article (what does this mean?). If published, this will include your full peer review and any attached files.

Reviewer #1: No

Reviewer #2: No

---

## [Author Response · Author response to Decision Letter 0]

28 Dec 2019

Daniel E. Duplisea, PhD 

Fisheries and Oceans Canada 

850 Route de la Mer 

Mont‐Joli, Quebec 

G5H 3Z4, Canada

2019-12-29

Dear Dr. Duplisea,

Thank you for considering a revision of the manuscript PONE-D-19-26904,

**“`SimSurvey`: an `R` package to optimize the design and analysis of

fisheries surveys by simulating spatially-correlated fish stocks”** by

Paul M. Regular, Gregory J. Robertson, Keith P. Lewis, Jonathan Babyn,

Brian Healey and Fran Mowbray. In accordance to one of the reviewers’

suggestions, the paper has been renamed **“`SimSurvey`: an `R` package

for comparing the design and analysis of fisheries surveys by simulating

spatially-correlated fish stocks”**.

We thank the Reviewers for their thoughtful and thorough comments which

will clarify and improve the manuscript. We agree with many of the

comments and have made every effort to incorporate these into the

manuscript. Below are the reviewers’ comments with a brief accounting of

our response to each concern or suggestion.

We submit this revised manuscript for your consideration and look

forward to your decision.

Sincerely,

Paul Regular 

Fisheries and Oceans Canada 

Northwest Atlantic Fisheries Center 

80 East White Hills, St. John’s, NL 

A1C 5X1, Canada 

E-mail:

Paul.Regular@dfo-mpo.gc.ca

Phone: (709) 772-2067

Reviewer \\#1:

This paper describes the R package “SimSurvey” which is a package to

test the design and analysis of fisheries independent surveys.

I found the paper very interesting and the package of great potential

value and interest to many fishery scientists. However, I have several

comments that I would like the authors to address first before I

recommend acceptance of this paper. My main comments relate to:

1. Verify some of the equations and model description used in the

 model. I think that some of them lack description to fully

 understand what was done and few others are misleading, even

 incorrect. The detail can be found below but in general, they are

 linked to the problem of bias correction for log-normal

 distribution, re-scaling”, moving from abundance at age to abundance

 at length.

2. Add additional explanation to justify the choices. E.g. why

 population dynamics is modeled independently of the spatial

 distribution. Is this reasonable, limitation, etc.

3. Few suggestions (add-ons, renaming) to increase the generality of

 the paper (without requiring too much work)

*We hope we have addressed each of these main comments by a) clarifying

several equations, b) adding more details and justifications, and c)

modifying naming conventions and clarifying extensibility. See replies

below for more details.*

Detailed comments:

Abstract:

What about the flexibility to include new sampling strategies (user

defined one) or including new estimation approaches

*We have noted that both built-in and user-defined sampling strategies

can be utilised.*

Introduction:

L43: quality of

*We have made the suggested change.*

It would be good to add few references here e.g. L50,

*We have added references as suggested.*

Method:

L69-73: I think it would be good to add the reference to the specific

function name listed in Table2. This way, I would be more explicit and

avoid possible confusion. For example, in my case, when I first read the

section header “simulate abundance”, I was expecting to see the way you

dealt with the spatial distribution of abundance.

*We have made the suggested change.*

L94&95&104: I just want to make sure that the bias correction has been

applied to these two equations. If logN (or logZ) is normally

distribution with mean mu and sigma, then the N (or Z) is lognormally

distributed with mean exp(mu+sigma^2/2). Therefore, when you write

“mean” in table 3 for R and Z, I hope that you modified in your actual

equation for log(Z) to: log(Z) ~ normal(log(mean)-sigma^2/2, sigma) In

other word, the mean(R) is not simply exp(meanR\\_log\\_scale).

exp(meanR\\_log\\_scale) is the median of the distribution of R.

*Good catch. The answer to the question is no, bias correction was not

applied, but nor was it needed, as all demographic parameters operated

in log space. The issue is a matter of naming convention that we

admittedly struggled with. When developing the functions, we vacillated

on whether to use `log_mean` or `mean` as the argument name and we

landed on mean as we believed this would be a more relatable value for

users to supply. To be more explicit with regards to what value should

be supplied, we have renamed the arguments to `log_mean`.*

L107: there is not enough description to understand how you converted

the abundance at age to abundance at length. What were the choice of

length bin groups? Every 5 cm, 10cm? Such information is also missing in

the tables. Please add that. But then you have to calculate some

cumulative distribution (until length l) for the length at age

distribution (using the VB growth equation) and do some subtractions to

be able to allocate the abundance at age to each length group. See

equation A.1.14 in here for example.

https://www.nwfsc.noaa.gov/news/events/program_reviews/documents/C.2_Methot_Wetzel_SSTechnicalDescription.pdf

*Another good catch as we did not provide enough information to

replicate the approach. We have added more detail on the calculations

behind the sim\\_vonB function; turns out our approach was similar to the

one described in the exemplar the reviewer provided.*

“Simulate spatial distribution”:

I have a few comments on this section:

1. First of all, I think it is important to state that you assume that

 the population dynamic model and the spatial distribution is

 independent i.e. you do not have a spatially-explicit population

 dynamics model. And talk about what it means (is it realistic,

 limitation, etc).

*We have prefaced this section with the caveat that population and

spatial dynamics are modeled as independent processes and we note the

pros and cons of this approach. In short: the con is that the approach

is a simplification of reality as it does not explicitly account for

dynamics such as larval dispersal, spatial differences in growth and

population connectivity; the pro is that this simplification limits the

number of unknown parameters that need to be specified while

facilitating the simulation of a sufficiently complex population for

testing the efficacy of various survey designs.*

1. L161. I think it would be better to rename “depth” as “main

 covariate influencing the species distribution”. And “depth” happens

 to be one of them for many species, but there are cases when this is

 not the case.

*A good point, this variable could be used for any habitat gradient over

which stratification is applied. However, we retain the term “depth” for

three reasons: 1) it is an important covariate for many species (as the

reviewer noted), 2) most surveys that we are aware of are

depth-stratified (also noted by the reviewer below) and 3) it is a

concise variable name (as opposed to something like “habitat\\_covar”).

We therefore retained the “depth” naming as it ought to satisfy most use

cases. We now note directly in the text that any covariate could be

supplied and used as ‘depth’*

1. L161. I think it would be good to say upfront that there are two

 ways of defining the spatial structure. Using the function in the

 package or user defined.

*Given a suggestion from Reviewer \\#2, we have restructured the paper to

have two core sections: “Model structure” and “Using **`SimSurvey`**”.

We have included a blanket statement under the “Model structure” section

noting that users can circumvent specific components of the

**`SimSurvey`** framework and supply their own objects, including

spatial structure of the survey grid.*

1. L161: I think another possibility is to generate a random field by

 using the package “Randomfields” for example. This way you can

 generate a map where “depth” is patchily distributed (maybe more

 like an island type case study)

*Good idea! We actually pursued this idea in an earlier iteration of

`make_grid`, however, we abandoned the option because it was difficult

to automate because the random field sometimes created small strata (one

cell) or strata that were split in space. We would certainly welcome

more development integrating other packages with **`SimSurvey`***

1. L163-164: please add more description about the division or strata.

 How can we set it up and what do you specifically mean by division

 and strata? Is one nested within the other, or not necessarily? Your

 examples are based on Atlantic Ocean and people in other regions

 might not be familiar with how these divisions are created.

*We have added more detail on the structure of the divisions and strata.

In short, divisions are pre-defined areas, generally within which a

fisheries management framework is applied. Age-length keys are specified

at the level of the division, so for populations spanning large habitat

gradients (in temperature for example) in which age-length relationships

change along that gradient, multiple divisions can be supplied. Strata,

on the other hand, are defined by an important habitat gradient. As

structured in **`SimSurvey`**, strata are nested in divisions. *

1. L163: Why only focus on “depth-based strata”? I think it would be

 good to allow the user to choose their own stratification approach.

 It could be depth based as you did (which happens most often in

 surveys) but it could technically be any other thing (user

 supplied). This allows more flexibility.

*We agree, see reply to point 2. above, and note that any habitat

variable could be used for stratification.*

1. L167: the equation is misleading and I am not sure it is right based

 on what is written in the text. You mentioned later on L178: that

 you “re-scaled” so that the total number of fish in a specific year

 and age across space is equal to the single number from the

 population dynamics model. If so, the re-scaling should be done in

 the identify (natural) scale, not in log scale. In log scale, even

 if ,, sums to zero across space, the sum in the identity scale won’t

 match. This is often refer to as “bias correction” for the log

 normal models. And you should ideally show how the rescaling was

 done in terms of the equations too.

*Right, the equation presented was not an accurate reflection of the

calculations. We have revised the equation to explicitly show how the

values were normalized to sum to 1.*

1. L167: this “depth preference” function is very simplistic and gives

 only very “smooth” symmetric distribution. More often, fish have a

 skewed depth preference: often right-skewed.

*True. Some users may find this parameterization insufficient for their

species and we hope they will implement their own closure to use in the

`sim_distribution` function to better simulate the effect. In addition

to our blanket statement under the “Model structure” section, we have

added a more specific statement under the “Using **`SimSurvey`**”

section stating that alternate formulations can be used by supplying

alternate closures to the core functions.*

1. L173-174: I think it might be worth adding, in simpler terms, the

 meaning of the spatial smoothing and scaling parameters.

*We have prefaced that sentence with “The rate at which point-to-point

spatial correlation decays with distance is controlled by…”.*

1. L178: it is another question of scaling. How did you exactly do the

 scaling? In the identity scale? By dividing my the sum of the

 effects? I am asking this because depending on how you did the

 rescaling, your correlation structure in space and age might have

 been affected and is not the same as the one specified in L172. Did

 you verify that?

*See reply to point 7. above*

Table 4. “group\\_ages”. Ok but how is the variance controlled for the

other age classes?

*“Variance” was a poor word choice. We have replaced it with “noise” as

it is the simulated noise that we fix across multiple age groups.*

L189: “user supplied”. This is a good feature. However, I think it is

important to mention here that user have to make sure that they use the

correct projection method to ensure that each grid is of the same

dimension.

*We have made the suggested change.*

L216: reference to figure 3?

*We have made the suggested change.*

L221: there is not “group\\_years” argument in Table 4.

*We have added it to the table.*

L237: “this function”. I think it would be better to replace “this

function” with “sim\\_distribution” as you do not mention the word

“sim\\_distribution” in the sentence above this.

*We have made the suggested change.*

L158-251: In general, I think re-organising this section using

sub-section headers could be useful. Just to guide the readers

*This is an excellent suggestion. By following a suggestion by Reviewer

\\#2 to re-organize the paper into two core sections we have added more

headers to help guide the readers.*

L254: you say that sampling is stratified random but SRS is also an

option based on Table5. Please correct.

*We have made the correction.*

L257: what does this mean? Does this control the number of set but how

is this calculated?

*We have clarified how number of sets per strata is calculated.*

L257: I do not see how you control for the total number of set in the

survey? How do you control it?

*We have clarified how number of sets per strata is calculated; it is

done within the `sim_survey` function using the `set_den` argument, with

further adjustment possible with the `min_sets` argument.*

L261: I think you should mention here that you can also force the sample

size (as seen on table 5). Moreover, in table 5, it would be good to

set-up a “ages\\_min” for the minimum number of ages to sample \\[…\\]” so

that it gives the ability to fix the sample size if needed by writing

the same value for “ages\\_min” and “ages\\_cap”.

*We are not sure what the reviewer would like to have implemented here.

Is the suggestion to impose a minimum number of ages to collect across

all length groups? In practice, an `ages_min` constraint is already

applied for length bins where the total number of fish caught is lower

than `ages_cap`. A specific `ages_min` constraint cannot really be

applied, as if there are too few fish of a certain size caught in any

one survey, there is no means of getting more fish of that size.*

L261: How are you making sure that the number of sampled fish for that

specific cell, age and year won’t be above the total number of fish in

that cell, age, and year? The probability value could be close to one

and if you fish in a few a time, then you are at risk. Especially

because your population dynamics model is not spatially explicit and is

completely independent of the distribution function itself i.e. you can

technically fish out all the fish in an area but it will be populated

back the year after the way you implemented in this study… Maybe you

need to put a condition (or just a note) for general users to make sure

that this probability value is much below 1?

*The sampling is implemented such that the number of fish sampled in a

cell cannot exceed the number of fish in a cell because the population

is split across sets in cases where more than one set is conducted in a

cell. We have added this missing detail to our manuscript. We also added

a note that the survey is assumed to have no impact on the population

from one year to the next, and users planning to conduct surveys at

levels that will affect the overall population will need to use caution

in interpreting their results.*

L267: I recommend to clarify something here. 1. Depending on the number

of fish caught? What do you mean? What is the rule you used? 2. The way

you coded, sample by age is first decided, then the corresponding length

is calculated, then age-subsample is determined. In reality, length

sample is taken in the field, then age sub-samples are taken. While

similar, I do not think it always equal. Especially, when you start

including some correlation structure in the sampling. By the way, did

you consider including some correlation structure in the sampling

process to make more realistic?

*Our wording “depending on the number of fish caught” was unfortunately

vague and, in hindsight, is not adding anything to the paper. Therefore,

we have removed the statement to minimize confusion. We have also

clarified the sub-sampling sequence. Finally, we have yet to consider

including correlation structure in the sampling process as we went about

imposing correlation via the spatial correlation of age groups

(i.e. age-specific clustering tends to result in sets with high

intraclass correlation). Correlation in the sampling procedure could be

a future development for **`SimSurvey`**, but we feel at this time the

package is sufficiently complex and we did not want to overwhelm users

with too many options.*

L275 Table 1 on should be table 5

*We have changed the page numbers accordingly.*

L275: Table5: “age\\_sammpling” should be “age\\_sampling”

*We have made the suggested change.*

L275: “min\\_sets” you have not described it yet and what is it? You have

sample from all cells? If no, this is not realistic.

*We have clarified the meaning and utility of the `min_sets` argument

(i.e. a small strata may be allocated only one set under a low set

density scenario; this argument overrides the allocation and imposes the

`min_sets` if it is greater than the allocation).*

L279: Table5 not Table1?

*We have changed the numbers accordingly.*

L285-286: Could you be more specific on how custom closures can be

supplied and where?

*We have included an example in the second paragraph of the “Using

**`SimSurvey`**” section that we hope will clarify how a user can supply

a custom closure.*

L306: how are these catchability corrected abundance matrices

calculated? It is important to write this information somewhere (or

write “please refer to the section “Stratified analysis” for further

information on the calculation of abundance indices”) or something alike

and Appendix S3.

*We have clarified how this was calculated.*

L336: I think it would be good to say that other methods exist and

people can use it in this package (maybe)?

*Good point, however, we think this is covered by referencing a paper

that describes a geostatistical R package and we also note that other

options can be used under the “Research opportunities” section.*

L421: color gradient. Even though it is obvious it might be good to say

green to purple gradient.

*We have made the suggested change.*

L427: instead of “sampling protocol”, I think it would be more meaning

full to say the maximum number of length samples.

*We have made the suggested change.*

L452: say that the color ramps from yellow to purple

*We have made the suggested change.*

S1 appendix: missing figure in S1

*We have included the figure*

Reviewer \\#2:

This manuscript describes an R package called SimSurvey. The package

includes a set of functions for simulating point-based fisheries survey

designs, e.g. bottom trawl surveys, for estimating abundance indices. It

focuses on number of stations and number of fish sampled ignores other

constraints such as distance between stations and day-time duration

which impose strong constraints on real surveys. The functions included

allow the user to first simulate age-and length structured population

dynamics, distribute individuals randomly in space (assuming a certain

correlation structure), carry out a survey and finally calculate

abundance indices from the simulated data. The package will likely be of

interest to researchers wanting to explore the precision achievable with

different survey designs. My comments regarding the package and the

presentation are summarised below.

1. Optimization

The title announces a package for optimizing survey designs. As far as I

can see the package does not allow survey design optimization, neither

in terms of defining survey strata nor in terms of number of stations

per stratum. The strata are defined by the user. The only option

available for the number of stations is proportional to stratum surface;

the user sets the minimum number of stations taken in the smallest

stratum. It would be useful to be able to specify the total number of

stations and test different allocation schemes, such as proportional to

surface area (implemented), equal number per stratum, Neyman allocation

(accounting for surface area and abundance variability), etc. Please

consider revising the title (e.g. “compare” and instead of “optimize)”

and spell out the available sampling design options.

*We agree with the Reviewer and retitled the ms “`SimSurvey`: an `R`

package for comparing the design and analysis of fisheries surveys by

simulating spatially-correlated fish stocks”. Currently the package

provides options for stratified random sampling proportional to surface

area, but other survey designs could added in future versions of

**`SimSurvey`**.*

1. Manuscript structure

The manuscript might be easier to follow if the manuscript was

restructured: 1) Model description, 2) Using SimSurvey. The later

section would then group all example code which could again be

subdivided into running simulations and exploring results (plot

functions).

*This is an excellent suggestion! We have re-structured our manuscript

accordingly and feel that this structure will be much easier for a

reader to follow.*

1. Parameterisation To use SimSurvey for a real world problem realistic

 parameter values are needed. The package comes with default values

 chosen for a particular case study. However, no mention is made in

 the manuscript how to choose appropriate values for the many model

 parameters to tailor the simulations to a population of interest. I

 suggest the authors add a section on parameterisation and a table

 summarizing all parameters with a column specifying how to

 parameterize. For example, parameters for population dynamics and

 growth could probably be taken from the literature (or a stock

 assessment report). However this is not possible for the parameters

 of the spatial distribution function sim\\_distribution() such as

 correlation between ages etc. Ideally the package would include a

 fitting function for estimating these parameters from actual data.

 These input data could come from a pilot survey and include location

 (lat, long) and numbers by length/age.

*This is yet another very helpful suggestion and, now that we have

included a Parameterisation section, it is easy to see the value of such

a section to prospective users of the package. Instead of a table that

summarizes how to specify the parameters, we wrote an entire section

that outlines recommended steps for setting up a their own simulation.

We started drafting a table, however, it quickly became apparent that it

would be too big and cumbersome and somewhat redundant with the core

tables that describe the arguments and parameters. We hope that the

narrative/outline we included will serve the practical purpose the

reviewer had in mind.*

Minor issues - line 93: I assume there is an age plus but this needs to

be mentioned. Also, please specify how you set the initial numbers for

plus group ().

*We have now noted in the manuscript that a plus group is not modeled

explicitly as the number of ages can easily be extended to include

groups with zero fish. This choice simplifies the simulation, including

the setting of initial numbers which is done via exponential decay.

Further, the lack of a plus group is inconsequential for survey based

estimates of abundance at age.*

- line 121: I don’t understand the explanation of a closure. What do

 you mean by “return functions”? Do you mean it returns an object

 with different attributes?

*We have improved our explanation of a closure at the beginning of the

Using **`SimSurvey`** section*

- line 125: the number of right and left brackets is unbalanced,

 please check

*We have provided an improved description of a closure; the logic behind

this line of code should be clearer now*

- line 126 “This structure was chosen to avoid the repeated

 specifications of ages and years”. As far as I can see the example

 code only specifies years, not ages.

*Again, we hope that our improved description of a closure will clarify

what we mean by this.*

- line 227 Please explain what a pipe is and how it is used. In the

 example I understand that the output of sim\\_abundance( ) is

 provided to (piped) into sim\\_distribution(). I am unclear what the

 object b contains. Is it the result of sim\\_distribution()?

*We have clarified how a pipe works, noting that it forwards values from

one function call to the next function call, and we now state that the

output from the two examples provided (nested approach vs. pipe

approach) are functionally the same though the approach is slightly

different.*

- There is no table 1, please revise table numbering.

*We have made the suggested change.*

---

## [Editor Report · Decision Letter 1]

6 Jan 2020

PONE-D-19-26904R1

SimSurvey: an R package for comparing the design and analysis of fisheries surveys by simulating spatially-correlated fish stocks

PLOS ONE

Dear Dr. Regular,

Thank you for submitting your manuscript to PLOS ONE. After careful consideration, we feel that it has merit but does not fully meet PLOS ONE’s publication criteria as it currently stands. Therefore, we invite you to submit a revised version of the manuscript that addresses the points raised during the review process.

**Please see the detailed suggestions below.**

We would appreciate receiving your revised manuscript by Feb 20 2020 11:59PM. To enhance the reproducibility of your results, we recommend that if applicable you deposit your laboratory protocols in protocols.io, where a protocol can be assigned its own identifier (DOI) such that it can be cited independently in the future. For instructions see: http://journals.plos.org/plosone/s/submission-guidelines#loc-laboratory-protocols

We look forward to receiving your revised manuscript.

Kind regards,

Daniel E. Duplisea, PhD

Academic Editor

PLOS ONE

Additional Editor Comments (if provided):

Dear Paul, you have made lots of excellent changes that I think help with the use of the package and responded to the many specific comments of reviewers which has no doubt corrected many technical issues and clarifications. The one thing I would say that you have not really done is get at the deeper research merits of this work beyond introducing a new piece of software. I think paper, as it stands, lacks a larger context and content which is important for the primary publication. My diagnosis for why this is is that the manuscript does not conform very well to more typical scientific reports (Intro, M&M, Result, Discussion) which can make it difficult for readers to find the larger scientific merits of the work. It is useful of course for those who already understand the merits of this kind of work but this work is for primary publication and it needs to appeal more to the former than the latter group. There is a very "how-to" feel to it (e.g. line 64 "In this section") which I think detracts from getting at the larger purpose of the work.

I would really like you to address this issue of moving it from a software manual to a primary scientific publication. I do not think it should involve that much work but there will be some restructuring of sections as well as places to put in content and bring out conclusions. Here are my suggestions for this:

Try to follow a more traditional paper structure. This will help readers and it likely will also make it clearer for you on how you can inject content into the paper to move it beyond the software manual approach:

Introduction:

You need to talk about wider issues and examples. e.g. examples of when poor survey design meant that scientific questions could not be properly addressed when good survey design meant they were. Examples of when good survey design allowed researchers to address needs that were unanticipated at the time it was designed. i.e. you need to build a better case (not just cost) of why survey design is really important and it is most powerful to do this with examples. You should try to bring in ideas related to ecosystem and climate changes and being able to track communities. Perhaps bring in species at risk ideas and tracking decline, you could bring in ideas related to MSC certification. These are just examples of specifics but you get the idea: the Introduction needs to have more general information outlining in both a broad and specific sense why we should be concerned about this.

Keep in mind an educated reader who may not be in fisheries but is interested in why anyone should care about this or could, for example, be interested in surveying say caribou or songbirds or something outside of fisheries but where many of the motivation and concepts may be similar and they are doing a more general literature search before designing their own survey.

Methods:

This will start at your "Model Structure" section. You should put a higher level heading just before that called Methods.

I can see that it is hard to separate your Methods from Results. Your results are really in the section "Using SimSurvey" I would not be opposed to putting this as a separate Results section but I leave it to you to decide. Essentially what you have is a case study as an example of how it works and therefore specific results are less interesting than how you got there with the package. You might title it something like "Results: running a SimSurvey simulation". This section also has a lot of content without a lot of explanation. for example, you have several large and complicated figures in a row. You should discuss not only what figures are meant to offer in the package but also what they mean. So for example on line 473 you refer to three figures (5,6,7) but you offer little interpretation of those figures just why you can make them which is another example of the limitations of the how-to manual approach. So you need to think of it as a case study and help someone decide the implications of their survey design.

Results:

see comment above

Discussion:

This starts just before "Research Opportunities". You can keep these sections but there should be more preamble before jumping right into research opportunities. I suggest a general paragraph(s) that segue into your subsections of "research opportunities", "future research". In the Discussion you should address again some of the broader issues from the Introduction, e.g. how could spatial (depth) distribution changes anticipated under climate change for some population be tackled by survey design exploration now so that we can continue to track these changing populations 20 years down the road and do not lose the signal. How can this software help with that?

I think something that can be useful for readers is to outline the steps in a thought process a researcher might undertake when setting up a survey (perhaps a separate subsection) and then how one might go about a SimSurvey run for this. It also gives you a good opportunity to discuss your multispecies ideas:What is your current problem/needsWhat are the constraints (HR and $ perspective)What are the constraints from a biological, physical perspectiveResources availableWhat are your anticipated future needs (i.e. if climate change is going to make cod go deeper, will you be able to track it in 20 years?)How could you consider some or all of these with SimSurvey

Your "Assumptions" section should come down into the Discussion

"Future Directions" should say something about randomfields re: Reviewer 1. Even if you just outline that you have considered it. You might also try to say something about optimisation of design which Reviewer 2 mentioned.

---

## [Author Response · Author response to Decision Letter 1]

20 Feb 2020

Additional Editor Comments:

Dear Paul, you have made lots of excellent changes that I think help with the use of the package and responded to the many specific comments of reviewers which has no doubt corrected many technical issues and clarifications. The one thing I would say that you have not really done is get at the deeper research merits of this work beyond introducing a new piece of software. I think paper, as it stands, lacks a larger context and content which is important for the primary publication. My diagnosis for why this is is that the manuscript does not conform very well to more typical scientific reports (Intro, M&M, Result, Discussion) which can make it difficult for readers to find the larger scientific merits of the work. It is useful of course for those who already understand the merits of this kind of work but this work is for primary publication and it needs to appeal more to the former than the latter group. There is a very "how-to" feel to it (e.g. line 64 "In this section") which I think detracts from getting at the larger purpose of the work.

I would really like you to address this issue of moving it from a software manual to a primary scientific publication. I do not think it should involve that much work but there will be some restructuring of sections as well as places to put in content and bring out conclusions. Here are my suggestions for this:

Try to follow a more traditional paper structure. This will help readers and it likely will also make it clearer for you on how you can inject content into the paper to move it beyond the software manual approach:

Introduction:

- You need to talk about wider issues and examples. e.g. examples of when poor survey design meant that scientific questions could not be properly addressed when good survey design meant they were. Examples of when good survey design allowed researchers to address needs that were unanticipated at the time it was designed. i.e. you need to build a better case (not just cost) of why survey design is really important and it is most powerful to do this with examples. You should try to bring in ideas related to ecosystem and climate changes and being able to track communities. Perhaps bring in species at risk ideas and tracking decline, you could bring in ideas related to MSC certification. These are just examples of specifics but you get the idea: the Introduction needs to have more general information outlining in both a broad and specific sense why we should be concerned about this.

- Keep in mind an educated reader who may not be in fisheries but is interested in why anyone should care about this or could, for example, be interested in surveying say caribou or songbirds or something outside of fisheries but where many of the motivation and concepts may be similar and they are doing a more general literature search before designing their own survey.

*Thank you for thinking beyond a fisheries biologist audience, as this is the audience we have been targeting from the onset. In hindsight we see that was short-sighted so we have modified our introduction, as suggested, to pitch the concept to a broader audience. The reason we did not think beyond fisheries biology is because we tend to have access to extensive age-based data while others (e.g. songbird and caribou biologists) do not. However, there are certainly other marine or terrestrial situations that could use our approach, the general framework we now present should be useful to adapt our work to systems we have not even considered.*

*Specifically, we have added an additional paragraph at the start of the paper discussing the importance of good survey design at a very general level. We also have removed references to fish and fisheries from some places in the abstract, to show our work can easily be generalized to other systems.*

Methods:

- This will start at your "Model Structure" section. You should put a higher level heading just before that called Methods.

- I can see that it is hard to separate your Methods from Results. Your results are really in the section "Using SimSurvey" I would not be opposed to putting this as a separate Results section but I leave it to you to decide. Essentially what you have is a case study as an example of how it works and therefore specific results are less interesting than how you got there with the package. You might title it something like "Results: running a SimSurvey simulation". This section also has a lot of content without a lot of explanation. for example, you have several large and complicated figures in a row. You should discuss not only what figures are meant to offer in the package but also what they mean. So for example on line 473 you refer to three figures (5,6,7) but you offer little interpretation of those figures just why you can make them which is another example of the limitations of the how-to manual approach. So you need to think of it as a case study and help someone decide the implications of their survey design.

*We really struggled to accommodate traditional Methods and Results sections because all of our re-structuring attempts defied the definitions of these sections. To maintain a logical flow that describes the how and why to use the package, we landed on a mix of methods and case study results/discussion through the body of the paper. We first describe the "Model structure", then the "Core functions", and then we describe and discuss the case study results in the "Interpretation" section.*

*The new "Interpretation" section is essentially a shortened version of the appendix on the case study, however, this iteration of the manuscript includes one new result. In the process of revising the paper and reading more literature, we stumbled upon a potential design-based solution to the bias introduced by the division-level age-length key (default approach). The design-based fix was easily implemented using the package and, after running the necessary simulations, we discovered that the resultant estimates appear to be unbiased. We feel this is an interesting and useful addition to the paper as it shows that the package can be used to identify issues and explore solutions.*

*We should also note that we have moved the "Parameterisation" section to an appendix. This was the section where we provide some guidance on how to modify default settings to suit specific needs. Following the addition of the "Interpretation" section, it became clear that this section no longer fits with the logical flow of the paper.*

*Overall, we feel that the structural changes we have made to the paper have minimized the how-to content, and the increased focus on the case study should bring more meaning to the complex figures shown through the paper and highlight the real-world implications of the package.*

*Below we discuss a number of our failed attempts to place the content into Methods and Results sections:*

*1) We tried to start the methods section with "Model Structure", as suggested, and shunt the figures to a "Results" section; this, however, broke the flow and left a gap in the methods section which held demonstrations of the core simulation functions but not the plotting functions.*

*2) We tried to keep the methods section as is and describe some of the case study results in the results section, however it seemed awkward to describe and discuss a solution to the age-length key problem in a results section.*

*3) Earlier iterations of the paper were focused entirely on the case study, the format was traditional and the package was a side-note in the methods section. It became apparent, however, that the package was more interesting and generally applicable than the case study results. We then began re-working the paper into a how-to format and, while moving away from a conventional format, we looked to several papers as a guide on how to document a R package (e.g. https://journals.plos.org/plosone/article?id=10.1371/journal.pone.0092725).*

Results:

- see comment above

*Please see reply to Methods comment.*

Discussion:

- This starts just before "Research Opportunities". You can keep these sections but there should be more preamble before jumping right into research opportunities. I suggest a general paragraph(s) that segue into your subsections of "research opportunities", "future research". In the Discussion you should address again some of the broader issues from the Introduction, e.g. how could spatial (depth) distribution changes anticipated under climate change for some population be tackled by survey design exploration now so that we can continue to track these changing populations 20 years down the road and do not lose the signal. How can this software help with that?

*In contrast to a Methods or Results section, a Discussion section was easier to accommodate and we have provided one. Here we lean on the case study results to reiterate the importance of planning a survey or testing an existing survey. We then use this preamble to segue into the "Research opportunities" section as suggested.*

- I think something that can be useful for readers is to outline the steps in a thought process a researcher might undertake when setting up a survey (perhaps a separate subsection) and then how one might go about a SimSurvey run for this. It also gives you a good opportunity to discuss your multispecies ideas:

 - What is your current problem/needs

 - What are the constraints (HR and $ perspective)

 - What are the constraints from a biological, physical perspective

 - Resources available

 - What are your anticipated future needs (i.e. if climate change is going to make cod go deeper, will you be able to track it in 20 years?)

 - How could you consider some or all of these with SimSurvey

*As much as we would like to address each of these points and work towards a survey design handbook, it would be hard to be as cohesive and comprehensive as Cochran (1977; "Sampling techniques") or Sutherland (2006; "Ecological census techniques: a handbook"). Moreover, we have yet to implement features that could help users assess some of these trade-offs (i.e. cost-benefit analysis). We are hopeful that these are questions that future versions of SimSurvey may be able to help address as more features are added to elevate it from a purely statistical toolbox to something that also considers the operational constraints of delivering a survey.*

- Your "Assumptions" section should come down into the Discussion

*Agreed, we have made the suggested change.*

- "Future Directions" should say something about randomfields re: Reviewer 1. Even if you just outline that you have considered it. You might also try to say something about optimisation of design which Reviewer 2 mentioned.

*Good point. We have noted in the "sim_distribution" section that a custom closure can be created that uses randomfields to simulate spatial noise. We have also noted in the "Future directions" section that we hope to implement alternatives to random or stratified random designs to allow for more comprehensive evaluations of various designs.*

---

## [Decision Letter · Decision Letter 2]

2 Apr 2020

PONE-D-19-26904R2

SimSurvey: an R package for comparing the design and analysis of surveys by simulating spatially-correlated populations

PLOS ONE

Dear Dr. Regular,

Thank you for submitting your manuscript to PLOS ONE. After careful consideration, we feel that it has merit but does not fully meet PLOS ONE’s publication criteria as it currently stands. Therefore, we invite you to submit a revised version of the manuscript that addresses the points raised during the review process.

Please address the reviewer's new comments.

We would appreciate receiving your revised manuscript by May 17 2020 11:59PM. To enhance the reproducibility of your results, we recommend that if applicable you deposit your laboratory protocols in protocols.io, where a protocol can be assigned its own identifier (DOI) such that it can be cited independently in the future. For instructions see: http://journals.plos.org/plosone/s/submission-guidelines#loc-laboratory-protocols

We look forward to receiving your revised manuscript.

Kind regards,

Daniel E. Duplisea, PhD

Academic Editor

PLOS ONE

Additional Editor Comments (if provided):

The reviewer has provided some additional comments which should be addressed.

Reviewer 2

Reviewer 2

Reviewers' comments:

Reviewer's Responses to Questions

**Comments to the Author**

1. If the authors have adequately addressed your comments raised in a previous round of review and you feel that this manuscript is now acceptable for publication, you may indicate that here to bypass the “Comments to the Author” section, enter your conflict of interest statement in the “Confidential to Editor” section, and submit your "Accept" recommendation.

Reviewer #1: All comments have been addressed

2. Is the manuscript technically sound, and do the data support the conclusions?

Reviewer #1: Yes

3. Has the statistical analysis been performed appropriately and rigorously? 

Reviewer #1: Yes

4. Have the authors made all data underlying the findings in their manuscript fully available?

Reviewer #1: Yes

5. Is the manuscript presented in an intelligible fashion and written in standard English?

Reviewer #1: Yes

6. Review Comments to the Author

Reviewer #1: This is my second time reviewing this manuscript and the authors did a great job addressing people's comments. Great job for that. The manuscript is now much clearer and read well, I think.

This time, I only have a few minor comments:

L19 : «there is a limited number of»

L103: Maybe it would be a good idea to justify why you are converting to abundance at length.

L240-242: I think it could be a good idea to show how to exactly do this (maybe obvious to advanced people but not to the general audience). This could be a simple reference to an example on the github repo.

Table 2: If bias correction was not implemented, I would suggest to change the labeling in the table here and elsewhere (if needed) and replace the “mean” by “median” (if the variable is transformed back to the original scale). If not, I would make clear that you talk about mean in log scale.

L326: I did not see how one can use INLA or RandomFields in Appendix S3. This is a similar comment as the one for line 240-242.

L481-490: the processing time is surprisingly long… it is not an issue in itself but this might refrain some people to using this package… I think some code optimization could help (e.g. vectorize, use matrix operation as much as possible, etc)

7. PLOS authors have the option to publish the peer review history of their article (what does this mean?). If published, this will include your full peer review and any attached files.

Reviewer #1: No

---

## [Author Response · Author response to Decision Letter 2]

16 Apr 2020

Reviewer #1: This is my second time reviewing this manuscript and the authors did a great job addressing people's comments. Great job for that. The manuscript is now much clearer and read well, I think.

*We thank the reviewer for the kind words.* 

This time, I only have a few minor comments: 

L19 : «there is a limited number of» 

*We have removed the 'the' that should not have been in that sentence.*

L103: Maybe it would be a good idea to justify why you are converting to abundance at length. 

*Agreed, we have started a new paragraph regarding abundance at length which begins: "In practice, abundance at age is often inferred from length data as it is easier to collect. Abundance at length is therefore simulated from abundance at age using the original von Bertalanffy growth curve"*

L240-242: I think it could be a good idea to show how to exactly do this (maybe obvious to advanced people but not to the general audience). This could be a simple reference to an example on the github repo. 

*Indeed, this is a helpful idea and we have now created a short vignette on creating closures and this can be accessed via the pkgdown site (https://paulregular.github.io/SimSurvey/articles/custom_closures.html). A link to this vignette has been added to the paper.*

Table 2: If bias correction was not implemented, I would suggest to change the labeling in the table here and elsewhere (if needed) and replace the “mean” by “median” (if the variable is transformed back to the original scale). If not, I would make clear that you talk about mean in log scale. 

*Good catch, we have added log after mean to ensure that readers know the functions are operating in log scale.*

L326: I did not see how one can use INLA or RandomFields in Appendix S3. This is a similar comment as the one for line 240-242. 

*We have added the following in parentheses: see the code behind `sim_ays_covar_sped` [https://github.com/PaulRegular/SimSurvey/blob/master/R/sim_dist_spde.R] for an example of how the `sim_ays_covar` closure was modified to apply a Stochastic Partial Differential Equation approach using the `INLA` package.*

L481-490: the processing time is surprisingly long… it is not an issue in itself but this might refrain some people to using this package… I think some code optimization could help (e.g. vectorize, use matrix operation as much as possible, etc) 

*We have stretched our programming skills to the limit to reduce the processing time as much as possible. Of course, said programming skills are largely self-taught, so there is surely room for improvement. While developing the package, we repeatedly profiled our code and vectorized as much as we could, leaning heavily on the `data.table` package. We have also applied parallel processing in places in an attempt to speed up the code. In other parts of the package, we learned that there is little we can do to reduce the processing time (e.g. Cholesky decomposition is a factor limiting the speed of the `sim_distribution` function). Another step to further optimize the code would be to translate some of the core functions to C++, however this is on the wish list and well outside the scope of the project at this point in time. We are hopeful that we have produced enough here to catalyze collaborations that help extend and optimize the package.*

---

## [Editor Report · Decision Letter 3]

23 Apr 2020

SimSurvey: an R package for comparing the design and analysis of surveys by simulating spatially-correlated populations

PONE-D-19-26904R3

Dear Dr. Regular,

We are pleased to inform you that your manuscript has been judged scientifically suitable for publication and will be formally accepted for publication once it complies with all outstanding technical requirements.

With kind regards,

Daniel E. Duplisea, PhD

Academic Editor

PLOS ONE

---

## [Editor Report · Acceptance letter]

27 Apr 2020

PONE-D-19-26904R3 

SimSurvey: an R package for comparing the design and analysis of surveys by simulating spatially-correlated populations 

Dear Dr. Regular:

I am pleased to inform you that your manuscript has been deemed suitable for publication in PLOS ONE. Congratulations! Your manuscript is now with our production department. 

With kind regards,

on behalf of

Dr Daniel E. Duplisea 

Academic Editor

PLOS ONE